# Lookahead Path Likelihood Optimization for Diffusion LLMs

Xuejie Liu [1 2 3]   Yap Vit Chun [4]   Yitao Liang [1]   Anji Liu [5]

## Abstract

Diffusion Large Language Models (dLLMs) support arbitrary-order generation, yet their inference performance critically depends on the unmasking order. Existing strategies rely on heuristics that greedily optimize local confidence, offering limited guidance for identifying unmasking paths that are globally consistent and accurate. To bridge this gap, we introduce *path log-likelihood* (Path LL), a trajectory-conditioned objective that strongly correlates with downstream accuracy and enables principled selection of unmasking paths. To optimize Path LL at inference time, we propose **POKE**, an efficient value estimator that predicts the expected future Path LL of a partial decoding trajectory. We then integrate this lookahead signal into **POKE-SMC**, a Sequential Monte Carlo-based search framework for dynamically identifying optimal unmasking paths. Extensive experiments across 6 reasoning tasks show that POKE-SMC consistently improves accuracy, achieving **2%–3%** average gains over strong decoding-time scaling baselines at comparable inference overhead on LLaDA models and advancing the accuracy–compute Pareto frontier.

## 1. Introduction

Recent advancements in Diffusion Large Language Models (dLLMs) (Nie et al., 2025; Bie et al., 2025; Ye et al., 2025; Gong et al., 2025) have introduced a compelling alternative to the dominant Autoregressive (AR) paradigm (Brown et al., 2020; Achiam et al., 2023). By framing text generation as a discrete denoising process, dLLMs break the strict left-to-right constraint of AR models, offering two primary promises: high-throughput parallel decoding and the flexibility of arbitrary-order generation. While the efficiency gains of parallel decoding have been well established (Song et al., 2025; Wu et al., 2025a; Labs et al., 2025; Wang et al., 2025c; Li et al., 2025), the potential of arbitrary-order generation remains comparatively underexplored.

Although dLLMs theoretically support any unmasking order, inference performance is critically sensitive to the specific decoding path (Kim et al., 2025; Ye et al., 2024). Existing unmasking strategies largely rely on heuristic rules (Ye et al., 2025; Nie et al., 2025). However, as shown in Figure 1 (Left), widely used heuristics perform inconsistently across different task domains, suggesting that optimal decoding orders are highly context-dependent and task-specific. This motivates a shift from static, one-size-fits-all heuristics toward *dynamic* unmasking strategies that can adaptively identify promising orders for each instance.

In this work, we bridge this gap by establishing the **Path Log-Likelihood (Path LL)**, the joint likelihood of a sequence specified by its unmasking order, as a reliable proxy for generation quality. Unlike metrics such as ELBO (Sahoo et al., 2024) or marginal path entropy (Lee et al., 2025; Chen et al., 2025), we empirically demonstrate in Figure 1 (Right) that Path LL exhibits a significantly stronger correlation with downstream task accuracy, making it an ideal objective for inference-time optimization.

Building on this observation, our goal is to *optimize the unmasking order to maximize Path LL* at inference time. The central challenge is that the space of possible orders is combinatorial, and evaluating the global potential of a partial sequence requires look-ahead, as early unmasking decisions can have long-range effects on the final Path LL. To address this, we propose **POKE** (**P**ath-**O**ptimistic **K**-step likelihood **E**stimator), an efficient value estimator that upper-bounds the expected future Path LL of a partial decoding state. Crucially, POKE is designed to remain accurate even under aggressive parallel rollouts, providing a calibrated, scalable guidance for trajectory-level search.

We further integrate POKE into the Sequential Monte Carlo framework (Doucet et al., 2001) and develop **POKE-SMC**, which enables value-guided search to prioritize high-quality unmasking paths dynamically. Empirically, we find that POKE-SMC achieves consistent accuracy improvements across reasoning benchmarks with favorable inference-time

[1]Institute for Artificial Intelligence, Peking University [2]School of Intelligence Science and Technology, Peking University [3]Ant Group [4]Department of Statistics and Data Science, National University of Singapore [5]School of Computing, National University of Singapore. Correspondence to: Anji Liu <anjiliu@nus.edu.sg>.

*Proceedings of the 43rd International Conference on Machine Learning*, Seoul, South Korea. PMLR 306, 2026. Copyright 2026 by the author(s).

scaling behavior: under matched inference budgets, our method achieves average accuracy gains of **3%–5%** on representative LLaDA models (Nie et al., 2025), while prior decoding-time scaling methods typically yield only **1%–2%** gains, and we consistently attains a superior accuracy–compute Pareto frontier compared to prior decoding-time scaling methods across varying inference budgets. (Wang et al., 2025b; Chen et al., 2025; Lee et al., 2025). **Our contributions are summarized as follows:**

- We establish **Path LL** as a superior proxy for generation quality in dLLMs, demonstrating its strong correlation with task accuracy.
- We propose **POKE**, a value estimator that efficiently performs lookahead evaluation to approximate the future Path LL.
- We introduce **POKE-SMC**, a value-guided search framework that adaptively optimizes unmasking orders. Across multiple tasks, our method consistently improves the accuracy–compute Pareto frontier.

**Conflict of Interest Disclosure.** This work received support from Ant Group, and one of the authors is an intern at Ant Group. The experiments use only publicly available models and benchmarks. The authors declare no other financial conflicts of interest.

## 2. Background

Diffusion Large Language Models (dLLMs) (Labs et al., 2025; Bie et al., 2025) frame text generation as the reverse of a stochastic corruption process. Unlike autoregressive models, dLLMs support non-autoregressive and parallel decoding by learning to recover original tokens from masked sequences (Austin et al., 2021a; Sahoo et al., 2024).

**Forward Process.** Given a clean sequence of $N$ categorical variables $\boldsymbol{x}^0 = (x_1, \ldots, x_N)$, the forward process progressively replaces tokens with the [MASK] symbol. At continuous time $s \in [0, 1]$, each token is independently masked with probability $s$, yielding a corrupted $\boldsymbol{x}^s$:

$$q_{s|0}(x_i^s \mid x_i^0) = \begin{cases} s, & \text{if } x_i^s = \text{[MASK]}, \\ 1-s, & \text{if } x_i^s = x_i^0, \\ 0, & \text{otherwise.} \end{cases} \quad (1)$$

**Training Objective.** dLLMs learn to denoise $\boldsymbol{x}^s$ by modeling the joint distribution of masked tokens $p_{\mathrm{D}}(\boldsymbol{x}_{\mathcal{M}_s} \mid \boldsymbol{x}_{\mathcal{O}_s}; \theta)$ as a product of independent marginals:

$$p_{\mathrm{D}}(\boldsymbol{x}_{\mathcal{M}_s} \mid \boldsymbol{x}_{\mathcal{O}_s}; \theta) = \prod_{i \in \mathcal{M}_s} p_\theta(x_i \mid \boldsymbol{x}_{\mathcal{O}_s}), \quad (2)$$

where $\mathcal{M}_s := \{i \mid x_i^s = \text{[MASK]}\}$ and $\mathcal{O}_s := \{i \mid x_i^s \neq \text{[MASK]}\}$ represent masked and observed token set, respectively. The model is optimized to maximize a variational

lower bound on the data log-likelihood (Ou et al., 2024):

$$\log p_\theta(\boldsymbol{x}^0) \geq \mathbb{E}_{s, \boldsymbol{x}^s} \left[ \frac{1}{s} \log p_{\mathrm{D}}(\boldsymbol{x}_{\mathcal{M}_s} \mid \boldsymbol{x}_{\mathcal{O}_s}; \theta) \right], \quad (3)$$

where $s \sim U(0, 1)$, $\boldsymbol{x}^s \sim q_{s|0}(\cdot \mid \boldsymbol{x}^0)$.

**Generation.** At inference time, the continuous time $s \in [0, 1]$ is discretized into $T$ steps via a monotonic schedule $\{s_t\}_{t=1}^T$, where larger $t$ corresponds to a higher masking ratio. At each step $t$, the model selects a subset of masked indices $\mathcal{Q}_t \subseteq \mathcal{M}_t$ to be decoded. Following Eq. 2, tokens within $\mathcal{Q}_t$ are sampled in parallel from the denoising distribution: $\boldsymbol{x}_{\mathcal{Q}_t} \sim p_{\mathrm{D}}(\boldsymbol{x}_{\mathcal{Q}_t} \mid \boldsymbol{x}_{\mathcal{O}_t}; \theta)$. By iteratively updating the observed context $\boldsymbol{x}_{\mathcal{O}_{t-1}} = \boldsymbol{x}_{\mathcal{O}_t} \cup \boldsymbol{x}_{\mathcal{Q}_t}$, the model refines the sequence from pure noise to structured text over $T$ steps.

**Unmasking Strategy.** While any unmasking order is theoretically supported by dLLMs due to their order-agnostic training objective, heuristic strategies are typically adopted in practice to ensure decoding quality (Huang et al., 2025a). A common principle for selecting $\mathcal{Q}_t$ is to prioritize positions with lower model-predicted uncertainty. For example, tokens with the highest maximum probability (confidence) (Nie et al., 2025) or the lowest predictive entropy (Ye et al., 2025) are often selected, both of which consistently outperform random ordering across various tasks. In contrast, margin-based samplers choose positions with the largest gap between the top two candidate probabilities (Kim et al., 2025), which is particularly effective for structured reasoning tasks such as Sudoku.

## 3. Are Heuristic Orders Sufficient?

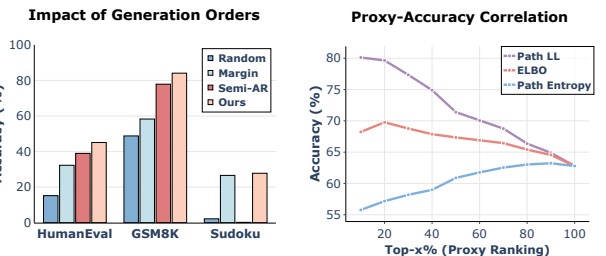

*Figure 1.* **What defines a good order?** *(Left)* **dLLM performance across diverse tasks using different ordering mechanisms.** No single static heuristic excels universally, whereas our method achieves consistent gains by dynamically optimizing Path LL. *(Right)* **Comparison of proxy objectives on GSM8K.** When ranking samples by proxy scores, Path LL exhibits a much stronger correlation with accuracy than the ELBO or Path Entropy.

While the heuristic strategies mentioned above improve upon random sampling, they remain "static" rules that fail to generalize across diverse tasks. As illustrated in Figure 1 *(Left)*, we compare representative ordering mechanisms across three distinct domains: HumanEval (Chen,

2021) (code), GSM8K (Cobbe et al., 2021) (math), and Sudoku (Ye et al., 2024) (planning). We observe a significant *heuristic inconsistency*: for instance, semi-autoregressive (semi-AR) decoding with confidence-based selection (Nie et al., 2025) excels on code and math but collapses on Sudoku, while margin-based sampling (Kim et al., 2025) is well-suited for Sudoku yet suboptimal elsewhere.

The task-specific nature of these heuristics motivates the search for a **dynamic unmasking strategy** that can adaptively identify good orders for each specific instance. However, such a search requires a robust objective to evaluate the quality of different decoding paths. For dLLMs, the training ELBO (Eq. 3) might seem like a natural candidate; however, it is theoretically order-agnostic. Because inference performance relies heavily on the specific unmasking trajectory, the ELBO fails to distinguish between effective and ineffective orders. This limitation motivates the use of path log-likelihood, which explicitly captures the sequential dependencies of the decoding process.

Empirically, we find that the ELBO is a poor predictor of success. As shown in Figure 1 (*Right*), we evaluate several sequence-level proxies on GSM8K by generating multiple decoding trajectories for each question and ranking them. While the ELBO and path entropy (Chen et al., 2025) exhibit weak or inconsistent correlations with downstream accuracy, our proposed **path log-likelihood (Path LL)** exhibits a strong positive correlation. Specifically, ranking samples by Path LL consistently yields the most reliable accuracy curves. For more details of Figure 1, see Appendix E.1

Formally, we define a decoding trajectory $\tau = (\mathcal{Q}_T, \ldots, \mathcal{Q}_1)$ as a sequence of sets of token indices unmasked at each step, forming a partition of $\{1, \ldots, N\}$. Starting from a fully masked sequence $\boldsymbol{x}^T$, the Path LL of a sequence $\boldsymbol{x}$ under trajectory $\tau$ is:

$$\log p_{\mathrm{D}}(\boldsymbol{x}; \theta, \tau) = \sum_{t=1}^{T} \log p_{\mathrm{D}}(\boldsymbol{x}_{\mathcal{Q}_t} \mid \boldsymbol{x}_{\mathcal{O}_t}; \theta), \quad (4)$$

where $\mathcal{O}_t = \bigcup_{j=t+1}^{T} \mathcal{Q}_j$ (with $\mathcal{O}_T = \emptyset$) is the set of tokens observed at step $t$. Unlike the order-agnostic ELBO, Path LL explicitly accounts for the sequential dependencies introduced by the unmasking trajectory $\tau$, making it a robust objective for identifying "good" unmasking orders.

## 4. How to Optimize for Path LL?

Motivated by the strong correlation between Path LL and downstream performance (Figure 1), we aim to identify unmasking trajectories $\tau$ that maximize the Path LL. This objective effectively transforms dLLM decoding into a search problem over the space of unmasking trajectories to find those that induce globally high-quality generations.

**The Look-ahead Challenge.** The central challenge in this search lies in the *look-ahead evaluation* of partial sequences. Since dLLM tokens are highly interdependent, each local unmasking decision can have long-range effects on the final Path LL. A greedy, locally optimal decision (e.g., picking the most "confident" tokens) may prematurely commit the model to a low-probability region of the trajectory space, leading to a suboptimal final sequence. However, accurately calculating the true future potential of a partial sequence is intractable, as it is computationally equivalent to solving the full combinatorial ordering problem. This highlights the need for an efficient approximation mechanism that can estimate the *global potential* without an exhaustive search.

**Value-Guided Search via Random Rollouts.** To address this complexity, we seek a method to estimate the **expected future Path LL** of a partial sequence. Ideally, one would average over all possible future trajectories, but the factorial growth of the search space makes this impossible. We therefore require a lightweight proxy that avoids the circular dependency of needing an optimal policy to evaluate the current policy. This motivates the use of uniformly random rollouts as a computationally efficient estimator. By simulating random completions, we can approximate the average quality of the remaining path without training a separate value network. This approach is inspired by value-based search methods like Monte Carlo Tree Search (MCTS) (Kocsis & Szepesvári, 2006), where the quality of a state is assessed via stochastic simulations.

To ensure our evaluation is not biased toward any specific heuristic, we adopt **uniformly random rollouts** (Gelly & Silver, 2007). In our context, this means that starting from a partially unmasked sequence, we complete the remaining tokens using a random unmasking order. Prior theoretical work suggests that averaging over such random paths can reliably preserve the relative ranking of different actions under mild smoothness conditions (James et al., 2017), providing a robust estimate of a state's intrinsic quality without the overhead of a complex rollout policy.

Formally, we define the **state value** $V(\boldsymbol{x}_{\mathcal{O}})$ as the expected future Path LL obtained by completing the remaining masked positions $\mathcal{M}$ under a uniformly random unmasking order $\sigma \sim \mathrm{Unif}(\mathrm{Perm}(\mathcal{M}))$:

$$V(\boldsymbol{x}_{\mathcal{O}}) \coloneqq \mathbb{E}_{\sigma, \boldsymbol{x}_{\mathcal{M}}} \left[ \log p_{\mathrm{D}}(\boldsymbol{x}_{\mathcal{M}} \mid \boldsymbol{x}_{\mathcal{O}}; \theta, \sigma) \right], \quad (5)$$

where $\boldsymbol{x}_{\mathcal{M}}$ is sampled from the denoising distribution $p_{\mathrm{D}}(\cdot \mid \boldsymbol{x}_{\mathcal{O}}; \theta, \sigma)$. While $V(\boldsymbol{x}_{\mathcal{O}})$ captures the expected *global* contribution of a partial sequence, its exact computation is intractable due to the combinatorial space of permutations $\sigma$ and token realizations $\boldsymbol{x}_{\mathcal{M}}$. This motivates the development of an efficient estimator $\hat{V}$ to enable scalable value-guided decoding.

## 5. Efficient Value Estimation

As established in Section 3, value-guided decoding hinges on accurately estimating the state value $V(\boldsymbol{x}_{\mathcal{O}})$ with minimal computational overhead. While dLLMs enable parallel generation, their inherent independence assumption, i.e., predicting tokens as a product of marginals, fails to capture the complex inter-token dependencies of the model's true joint distribution. Consequently, naive Monte Carlo rollouts using parallel predictions often lead to severe underestimation of the future path LL (Liu et al., 2025a; Xu et al., 2025).

To formalize this, we distinguish between the model's joint distribution and its factorized approximation. Let $p_{\theta}(\mathbf{X}_{\mathcal{M}} \mid \boldsymbol{x}_{\mathcal{O}})$ be the joint distribution over the masked variables induced by the uniform mixture of all possible unmasking orders. Specifically, for any single permutation $\sigma$, the joint distribution is defined by the chain rule: $p_{\theta}(\boldsymbol{x}_{\mathcal{M}} \mid \boldsymbol{x}_{\mathcal{O}}, \sigma) = \prod_{k=1}^{|\mathcal{M}|} p_{\theta}(\boldsymbol{x}_{\sigma_k} \mid \boldsymbol{x}_{\sigma_{1:k-1}}, \boldsymbol{x}_{\mathcal{O}})$. The true joint distribution is then the average over all permutations:

$$p_{\theta}(\mathbf{X}_{\mathcal{M}} \mid \boldsymbol{x}_{\mathcal{O}}) \coloneqq \mathbb{E}_{\sigma}[p_{\theta}(\mathbf{X}_{\mathcal{M}} \mid \boldsymbol{x}_{\mathcal{O}}, \sigma)], \quad (6)$$

which is intractably to query from the model. In contrast, following Eq. 2, we define the factorized distribution $q_{\theta}(\mathbf{X}_{\mathcal{M}} \mid \boldsymbol{x}_{\mathcal{O}}) \coloneqq \prod_{i \in \mathcal{M}} p_{\theta}(X_i \mid \boldsymbol{x}_{\mathcal{O}})$, which can be easily computed in a single forward pass.

**Value Decomposition.** Notably, the state value $V(\boldsymbol{x}_{\mathcal{O}})$ defined in Eq. 5 is exactly the expected LL of the joint distribution: $V(\boldsymbol{x}_{\mathcal{O}}) = \mathbb{E}_{\boldsymbol{x}_{\mathcal{M}} \sim p_{\theta}}[\log p_{\theta}(\boldsymbol{x}_{\mathcal{M}} \mid \boldsymbol{x}_{\mathcal{O}})]$. By comparing this to the tractable $q_{\theta}$, we can decompose the value as:

$$V(\boldsymbol{x}_{\mathcal{O}}) = \underbrace{\mathbb{E}_{\boldsymbol{x}_{\mathcal{M}} \sim p_{\theta}}[\log q_{\theta}(\boldsymbol{x}_{\mathcal{M}} \mid \boldsymbol{x}_{\mathcal{O}})]}_{\text{Product LL}} + \text{TC}(\mathbf{X}_{\mathcal{M}} \mid \boldsymbol{x}_{\mathcal{O}}),$$
$$(7)$$

where the residual term $\text{TC}(\mathbf{X}_{\mathcal{M}} \mid \boldsymbol{x}_{\mathcal{O}})$ represents the information "lost" when assuming masked variables are independent. This gap is formally known as **Total Correlation**.

**Definition 5.1** (Total Correlation). *Conditioned on the observed context $\boldsymbol{x}_{\mathcal{O}}$ (omitted for brevity), the total correlation of masked variables $\mathbf{X}_{\mathcal{M}}$ is the KL-divergence between the joint distribution $p$ and the product of its marginals:*

$$\text{TC}(\mathbf{X}_{\mathcal{M}}) = D_{\text{KL}}\left(p(\mathbf{X}_{\mathcal{M}}) \,\middle\|\, \prod_{i \in \mathcal{M}} p(X_i)\right). \quad (8)$$

**$K$-stage Decomposition.** In practice, we consider an iterative $K$-stage unmasking procedure ($K \ll |\mathcal{M}|$) to balance estimation quality with inference speed. For a specific permutation $\sigma$, we partition the masked indices $\mathcal{M}$ into $K$ disjoint groups $\{\mathcal{A}_1, \ldots, \mathcal{A}_K\}$ of equal size

$|\mathcal{A}| = |\mathcal{M}|/K$. Defining the revealed context at stage $k$ as $\boldsymbol{x}_{\mathcal{O}_k} \coloneqq \boldsymbol{x}_{\mathcal{O} \cup \mathcal{A}_1 \cup \cdots \cup \mathcal{A}_{k-1}}$, the value decomposition in Eq. 7 generalizes additively across stages:

$$V(\boldsymbol{x}_{\mathcal{O}}) = \mathbb{E}_{p_{\theta}}\left[\sum_{k=1}^{K} \Big(\log q_{\theta}(\boldsymbol{x}_{\mathcal{A}_k} \mid \boldsymbol{x}_{\mathcal{O}_k}) + \text{TC}_k\Big)\right], \quad (9)$$

where $\text{TC}_k = \text{TC}(\mathbf{X}_{\mathcal{A}_k} \mid \boldsymbol{x}_{\mathcal{O}_k})$.

**Optimistic Approximation.** The remaining challenge is to estimate the cumulative total correlation in Eq. 9 without expensive queries. We seek an **optimistic upper bound** for this term. This choice is motivated by the principle of admissibility in search: by ensuring we never underestimate the true value, we guarantee that we do not wrongly discard partial sequences that are actually superior. This prevents the search from prematurely pruning trajectories that, despite low local scores, retain high global potential.

By leveraging the "pinning" analysis of Anari et al. (2025), we observe that randomizing the unmasking order effectively shatters" token dependencies. This allows us to bound the expected discrepancy using only the marginal entropies, which are already computed by the model.

**Lemma 5.2** (Cumulative Total Correlation Bound). *Let $\sigma \sim \text{Unif}(\text{Perm}(\mathcal{M}))$ be a uniformly random permutation inducing the partition $\{\mathcal{A}_1, \ldots, \mathcal{A}_K\}$ and context variables $\{\mathbf{X}_{\mathcal{O}_k}\}$. The expected cumulative total correlation satisfies:*

$$\mathbb{E}_{\sigma}\left[\sum_{k=1}^{K} \text{TC}(\mathbf{X}_{\mathcal{A}_k} \mid \mathbf{X}_{\mathcal{O}_k})\right] \leq \frac{1}{K} \sum_{i \in \mathcal{M}} \mathcal{H}(X_i \mid \boldsymbol{x}_{\mathcal{O}}), \quad (10)$$

*where $\mathcal{H}(X_i \mid \boldsymbol{x}_{\mathcal{O}})$ is the marginal entropy under the dLLM conditional distribution.*

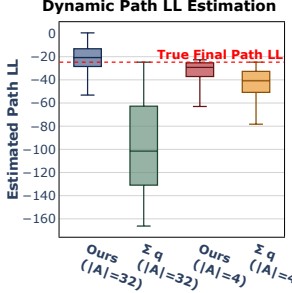

*Figure 2.* **High-Fidelity Path LL Estimation.** We aggregate path LL estimates across all intermediate steps for a GSM8K sample. While the product LL ($\sum \log q_{\theta}$) severely underestimates the true LL under large $|\mathcal{A}|$ (small $K$), POKE effectively bridges this gap and achieves accurate estimates.

**The POKE Estimator.** Combining the $K$-stage decomposition (Eq. 9) with the pinning bound (Eq. 10), we propose the **P**ath-**O**ptimistic **K**-step LL **E**stimator (**POKE**). By approximating the expectation with $R$ independent random rollouts, the POKE estimator is defined as:

$$\hat{V}_{\text{POKE}}(\mathbf{x}_{\mathcal{O}}) \coloneqq \frac{1}{R} \sum_{r=1}^{R} \left[\sum_{k=1}^{K} \log q_{\theta}(\mathbf{x}_{\mathcal{A}_k^{(r)}} \mid \mathbf{x}_{\mathcal{O}_k^{(r)}})\right]$$
$$+ \frac{1}{K} \sum_{i \in \mathcal{M}} \mathcal{H}(X_i \mid \mathbf{x}_{\mathcal{O}}) \quad (11)$$

---

**Algorithm 1** POKE Value Estimation

---

1: **Input:** $\boldsymbol{x}_{\mathcal{O}}$, $\theta$, rollout steps $K$ and count $R$.
2: **Init:** $p_\theta(\cdot \mid \boldsymbol{x}_{\mathcal{O}}), \{\mathcal{H}_i\}_{i \in \mathcal{M}} \leftarrow \text{ModelForward}(\boldsymbol{x}_{\mathcal{O}})$
3: $V_{\text{total}} \leftarrow 0$
4: **for** $r = 1 \ldots R$ **in parallel do**
5:     Partition $\mathcal{M}$ into $K$ random subsets $\{\mathcal{A}_1, \ldots, \mathcal{A}_K\}$
6:     $\mathcal{O}_1 \leftarrow \mathcal{O}$
7:     **for** $k = 1 \ldots K - 1$ **do**
8:         Sample $\boldsymbol{x}_{\mathcal{A}_k} \sim p_\theta(\cdot \mid \boldsymbol{x}_{\mathcal{O}_k})$
9:         $V_{\text{total}} \leftarrow V_{\text{total}} + \log q(\boldsymbol{x}_{\mathcal{A}_k} \mid \boldsymbol{x}_{\mathcal{O}_k})$
10:         $\boldsymbol{x}_{\mathcal{O}_{k+1}} \leftarrow \boldsymbol{x}_{\mathcal{O}_k} \cup \boldsymbol{x}_{\mathcal{A}_k}$
11:         $p_\theta(\cdot \mid \boldsymbol{x}_{\mathcal{O}_{k+1}}) \leftarrow \text{ModelForward}(\boldsymbol{x}_{\mathcal{O}_{k+1}})$
12:     **end for**
13: **end for**
14: $V_{\text{base}} \leftarrow V_{\text{total}}/R$
15: $V_{\text{corr}} \leftarrow \frac{1}{K} \sum_{i \in \mathcal{M}} \mathcal{H}_i$
16: **Return:** $\hat{V}_{\text{POKE}} = V_{\text{base}} + V_{\text{corr}}$

---

Notably, POKE counteracts the independence bias of parallel rollouts via the entropy correction, which naturally vanishes at a rate of $1/K$ as the decoding becomes more sequential. This mechanism is empirically validated in Figure 2, where we evaluate the estimated Path LL, i.e., the sum of realized LL and the estimated future value, across all intermediate steps of a GSM8K sample. (see Appendix E.2)

The results demonstrate that while the naive product LL ($\sum \log q_\theta$) severely underestimates the true joint LL as parallelization increases (e.g., unmasking $|\mathcal{A}| = 32$ tokens per step), POKE remains remarkably consistent with the oracle. The detailed algorithm is presented in Algorithm 1. For a formal proof of the POKE estimator, see Section F.

## 6. Path LL Optimization via SMC Search

Our primary goal is to identify a decoding trajectory $\tau$ that maximizes the path LL (Eq. 4). In contrast to autoregressive decoding, dLLMs require joint reasoning over token values and unmasking orders, leading to a combinatorially large search space. We address this challenge with **POKE-SMC**, which integrates our lookahead value estimator into a Sequential Monte Carlo (SMC) framework (Doucet et al., 2001) to efficiently explore high-quality decoding paths.

POKE-SMC maintains a population of $M$ particles that evolve from fully masked to fully decoded. While the underlying SMC machinery follows standard practice (Doucet et al., 2001), our method introduces several key design choices that are crucial for effective search in dLLMs:

**Global-Guided Weighting.** To accurately compare particles that may follow different unmasking orders, we define importance weights based on the estimated total path log-likelihood. This ensures that all particles are evaluated on

---

**Algorithm 2** POKE-SMC Search

---

**Require:** dLLM $p_\theta$, total steps $T$, particle count $M$, resampling interval $\Delta$, resampling temperature $\tau_r$, POKE rollout steps $K$.
1: **Initialization**: Initialize $\{\boldsymbol{x}_{\mathcal{O}_T}^{(m)}\}_{m=1}^M$ with $\mathcal{O}_T = \emptyset$ and $L_T^{(m)} = 0$ for all $m \in [M]$.
2: **for** $t = T \ldots 1$ **do**
3:     **// 1. Path Proposal**
4:     Select $\mathcal{Q}_t$ and
5:     **for each** $m$ **in parallel do**
6:         $\boldsymbol{x}_{\mathcal{Q}_t}^{(m)} \sim p_{\text{D}}(\cdot \mid \boldsymbol{x}_{\mathcal{O}_t}^{(m)}; \theta)$.
7:         $\boldsymbol{x}_{\mathcal{O}_{t-1}}^{(m)} \leftarrow \boldsymbol{x}_{\mathcal{O}_t}^{(m)} \cup \boldsymbol{x}_{\mathcal{Q}_t}^{(m)}$.
8:         $L_{t-1}^{(m)} \leftarrow L_t^{(m)} + \log p_{\text{D}}(\boldsymbol{x}_{\mathcal{Q}_t}^{(m)} \mid \boldsymbol{x}_{\mathcal{O}_t}^{(m)}; \theta)$.
9:     **end for**
10:     **// 2. Global-Guided Weighting & Resampling**
11:     **if** $(T - t + 1) \pmod{\Delta} = 0$ **or** $t = 1$ **then**
12:         **for each** particle $m$ **in parallel do**
13:             Compute $\hat{V}_{\text{POKE}}(\mathbf{x}_{\mathcal{O}_{t-1}}^{(m)})$ via Algorithm 1
14:             $\log w_t^{(m)} \leftarrow L_{t-1}^{(m)} + \hat{V}_{\text{POKE}}(\boldsymbol{x}_{\mathcal{O}_{t-1}}^{(m)})$
15:         **end for**
16:         Normalize: $\bar{w}_t^{(m)} \leftarrow \frac{\exp(\log w_t^{(m)}/\tau_r)}{\sum_{j=1}^M \exp(\log w_t^{(j)}/\tau_r)}$
17:         Resample $M$ particles from $\{\mathbf{x}_{\mathcal{O}_{t-1}}^{(m)}, L_{t-1}^{(m)}\}_{m=1}^M$ according to $\{\bar{w}_t^{(m)}\}_{m=1}^M$
18:     **end if**
19: **end for**
20: **Output**: Return $\boldsymbol{x}^{(m)}$ with the highest $L_0^{(m)}$.

---

a uniform global scale regardless of which specific tokens they have unmasked so far. For a particle $m$ at a resampling step $t$, we first compute its accumulated path LL:

$$L_t^{(m)} = \sum_{j=T}^t \log p_{\text{D}}(\boldsymbol{x}_{\mathcal{Q}_j}^{(m)} \mid \boldsymbol{x}_{\mathcal{O}_j}^{(m)}; \theta). \tag{12}$$

The total importance weight $w_t^{(m)}$ is constructed by augmenting Eq. 12 with $\hat{V}_{\text{POKE}}$ (the future path LL estimate):

$$\log w_t^{(m)} = \left( L_t^{(m)} + \hat{V}_{\text{POKE}}(\boldsymbol{x}_{\mathcal{O}_{t-1}}^{(m)}) \right). \tag{13}$$

Here, $L_t^{(m)} + \hat{V}_{\text{POKE}}$ represents a high-fidelity estimate of the final log-likelihood of the complete sequence.

**Efficient Periodic Resampling.** To balance efficiency and accuracy, we perform resampling only at fixed intervals rather than at every decoding step. This design significantly reduces computational overhead while preserving the benefits of value-guided selection. The resampling temperature and interval are denoted as $\tau_r$ and $\Delta$, respectively.

After all tokens are unmasked, the particle with the highest terminal Path LL is returned as the decoded sequence. For the full SMC pipeline, see Algorithm 2 and Appendix A.

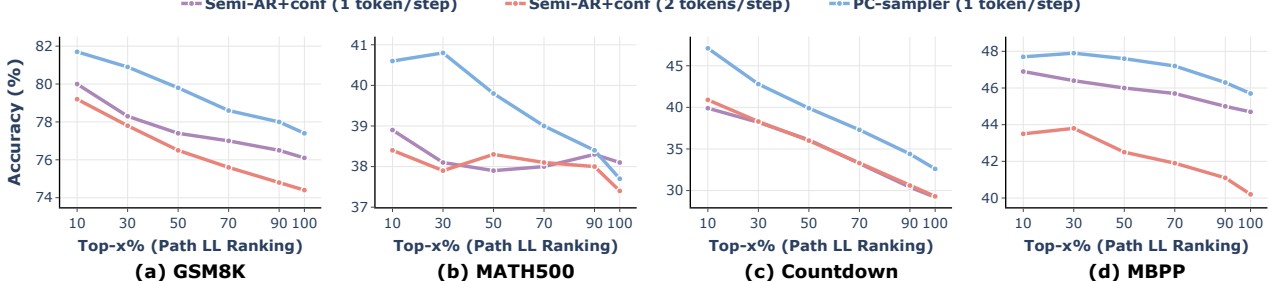

*Figure 3.* Accuracy of top-$x\%$ trajectories ranked by Path LL across four tasks (GSM8K, MATH500, Countdown, MBPP) and three decoding regimes. For each task, 400 problems are sampled with 10 trajectories each. The trend shows that trajectories with higher Path LL consistently correspond to higher accuracy, indicating that Path LL provides a reliable cross-task and cross-regime ranking signal.

# 7. Experiment

In this section, we empirically evaluate the effectiveness of the POKE-SMC framework. To provide a rigorous and comprehensive justification, our evaluation decomposes the framework into its core components and systematically addresses three critical dimensions: (i) **Fidelity of the Optimization Target**, examining whether the proposed Path LL serves as a reliable and generalizable proxy for downstream task accuracy (Section 7.2); (ii) **Accuracy of the Value Estimator**, validating whether the POKE estimator accurately approximates the true future path potential, preserves the sequence-level ranking oracle, and maintains computational efficiency (Appendix G); and (iii) **End-to-End Performance Scaling**, evaluating whether integrating this lookahead value guidance into a Sequential Monte Carlo (SMC) framework successfully boosts the generation quality and advances the accuracy-compute Pareto frontier of State-of-the-Art (SoTA) dLLMs (Section 7.3).

## 7.1. Experimental Setup

**Models and Benchmarks.** We conduct experiments on two representative dLLMs: **LLaDA-8B-Instruct** (Nie et al., 2025) and **LLaDA-1.5-8B** (Zhu et al., 2025) and evaluate POKE-SMC across a diverse suite of reasoning-intensive benchmarks: (i)*Mathematical reasoning:* GSM8K (Cobbe et al., 2021) and MATH500 (Lightman et al., 2023). (ii) *Code generation*: HumanEval (Chen, 2021) and MBPP (Austin et al., 2021b). (iii)*Planning:* Sudoku (Ye et al., 2024) and Countdown (Nolte et al., 2024).

**Baselines.** We compare POKE-SMC against several classes of dLLM sampling baselines: (i) *Heuristic Samplers:* Standard **Uniform** sampling (Austin et al., 2021a) and uncertainty-based methods (**Confidence** (Chang et al., 2022), **Entropy** (Ye et al., 2025), **Margin** (Kim et al., 2025)), and the efficient **EB-Sampler** (Entropy-Bounded) (Ben-Hamu et al., 2025). (ii) *Samplers with Positional Biases:* **Semi-AR** (Nie et al., 2025) (using 8 sequential blocks) with intra-block selection (**+Conf**, **+Ent**,

**+Marg**), and **PC-Sampler** (Huang et al., 2025a), which calibrates confidence via position-aware weighting and token frequency adjustments. (iii) *Inference Scaling Baselines:* To match the increased budget of POKE-SMC, we include: (1) **Majority Voting**, scaling via independent samples; (2) **ReMDM** (Wang et al., 2025b), which scales by adding iterative remasking and self-correction steps; and (3) **E-SMC** (Chen et al., 2025), a search-based baseline utilizing state entropy as guidance. E-SMC represents the class of entropy-guided search (similar to LookUM (Lee et al., 2025), see Appendix D for the baseline details);

**Scope of Evaluation.** While our end-to-end evaluations span all models and benchmarks, our component fidelity analyses (Section 7.2 and Appendix G) focus on four tasks using LLaDA-8B-Instruct to provide representative insights.

## 7.2. Generalizability and Fidelity of Path LL

Our framework optimizes the unmasking order under the foundational premise that the Path LL (Eq. 4) exhibits a strong correlation with downstream task accuracy. While this correlation was initially highlighted on GSM8K (Figure 1), we further examine its generalizability across broader scenarios. Specifically, we extend the Path LL fidelity analysis to four representative tasks (GSM8K, MATH500, Countdown, MBPP) across three decoding regimes (Semi-AR+conf with 1 token/step, Semi-AR+conf with 2 tokens/step, and PC-Sampler with 1 token/step).

For each problem, we sample 10 decoding trajectories, which are subsequently ranked based on their final Path LL scores. In Figure 3, we report the accuracy of the top-$x\%$ trajectories within each bucket. As illustrated in Figure 3, the empirical results demonstrate a remarkably consistent trend across all 4 benchmarks and all 3 decoding regimes: higher Path LL trajectories generally yield higher task accuracy. This predictive pattern manifests most prominently on GSM8K and Countdown, and remains clearly discernible for MBPP and MATH500 despite being noisier. These findings firmly validate Path LL as a robust, domain-agnostic,

*Table 1.* **Performance on LLaDA models across six benchmarks.** Each step decodes 1 token. The best score is highlighted in **bold**, and the second best is underlined. Among Multi-Sample inference baselines, our method achieves the most improvement under equal budgets, where $M$ denotes the number of particles. Results marked with † are taken from Huang et al. (2025a).

| Methods | HumanEval | MBPP | GSM8K | MATH500 | Countdown | Sudoku | Avg. |
|---|---|---|---|---|---|---|---|
| **LLaDA-8B-Instruct** | | | | | | | |
| Uniform† (2021a) | 15.2 | 24.6 | 48.8 | 23.0 | 14.4 | 2.2 | 21.4 |
| Confidence† (2022) | 27.4 | 42.4 | 59.1 | 25.8 | 34.0 | 23.8 | 35.4 |
| Entropy† (2025) | 28.1 | 42.2 | 60.9 | 23.6 | 33.8 | 1.6 | 31.7 |
| Margin† (2025) | 32.3 | 42.4 | 58.3 | 25.6 | 33.9 | 26.6 | 36.5 |
| EB-Sampler† (2025) | 26.8 | 43.3 | 61.2 | 23.8 | 34.1 | 24.2 | 35.6 |
| Semi-AR + Conf (2025) | 39.0 | 45.2 | 77.9 | 38.0 | 32.6 | 0.0 | 38.8 |
| Semi-AR + Ent | 40.2 | 45.4 | 80.6 | 37.8 | 31.5 | 0.2 | 39.3 |
| Semi-AR + Marg | 39.6 | 44.7 | 81.1 | 39.4 | 32.6 | 0.0 | 39.6 |
| ReMDM (2025b) | 32.6 | 34.0 | 77.9 | 25.4 | 34.0 | 23.8 | 38.0 |
| PC-sampler(2025a) | 42.7 | 46.6 | 79.2 | 38.4 | 36.3 | 25.8 | 44.8 |
| *Multi-Sample Inference Methods (based on PC-sampler)* | | | | | | | |
| +Majority Voting (M=5) | - | - | 81.4 | 40.0 | 36.9 | - | - |
| +E-SMC (M=5) (2025) | 41.5 | 45.0 | 81.9 | 39.0 | 40.4 | 25.6 | 45.6 0.8↑ |
| +POKE-SMC (M=4) | **45.1** | **47.1** | **84.1** | **41.0** | **40.8** | **27.8** | **47.7** 2.9↑ |
| **LLaDA-1.5-8B** | | | | | | | |
| Uniform† (2021a) | 17.7 | 23.0 | 52.7 | 21.2 | 15.8 | 3.4 | 22.3 |
| Confidence† (2022) | 28.1 | 43.3 | 60.7 | 24.4 | 33.8 | 24.8 | 35.9 |
| Entropy† (2025) | 32.9 | 44.0 | 60.3 | 23.6 | 34.7 | 0.2 | 32.6 |
| Margin† (2025) | 25.0 | 43.3 | 57.5 | 24.6 | 31.8 | 33.6 | 36.0 |
| EB-Sampler† (2025) | 32.9 | 43.6 | 61.1 | 24.0 | 34.6 | 0.2 | 32.7 |
| Semi-AR + Conf (2025) | 39.6 | 46.8 | 80.7 | 38.4 | 32.4 | 0.0 | 39.7 |
| Semi-AR + Ent | 37.8 | 46.6 | 81.2 | 39.2 | 30.9 | 0.2 | 39.3 |
| Semi-AR + Marg | 40.2 | 44.7 | 81.5 | 38.4 | 31.5 | 0.0 | 39.4 |
| ReMDM(2025b) | 33.8 | 40.5 | 80.1 | 24.4 | 33.8 | 24.8 | 39.6 |
| PC-sampler(2025a) | **45.7** | 49.9 | 80.5 | 42.2 | 34.9 | 28.8 | 47.0 |
| *Multi-Sample Inference Methods (based on PC-sampler)* | | | | | | | |
| +Majority Voting (M=5) | - | - | 83.3 | 41.8 | 36.0 | - | - |
| +E-SMC (M=5) (2025) | 42.7 | 48.0 | 83.3 | 42.2 | 38.6 | 28.8 | 47.3 0.3↑ |
| +POKE-SMC (M=4) | 45.1 | **50.4** | **83.6** | **44.0** | **39.7** | **31.4** | **49.0** 2.0↑ |

### 7.3. Overall Performance of POKE-SMC

Having established the fidelity of Path LL and POKE, we now focus on the end-to-end performance of the complete POKE-SMC framework. Crucially, standard dLLM heuristics reach a performance ceiling when decoding sequentially (i.e., one token per step). To achieve further gains, one must shift toward **inference-time scaling** and utilize additional computational budget via multiple particles or additional iterative remasking and correction. Consequently, our analysis emphasizes **budget-matched comparisons**: we evaluate whether POKE-SMC is more cost-effective than other scaling-oriented baselines, such as multi-sample voting/search or iterative remasking.

**Implementation Details.** POKE-SMC utilizes PC-Sampler as the base path proposal policy, a choice kept consistent for all multi-particle baselines (E-SMC and Majority Voting). To expand trajectories, we sample from the top-$k$ positions using a categorical distribution: $P(i \mid \mathbf{x}_{\mathcal{O}}) \propto \exp(s_i/\tau_p)$, where $s_i$ are scores from the base policy and $\tau_p$ is the proposal temperature.

The primary computational overhead of POKE-SMC stems from maintaining $M$ particles and performing POKE evaluations during resampling. We set $M = 4$ and a resampling interval $\Delta = T/4$. For POKE, we employ $R = 2$ rollouts with $K = |\mathcal{M}|/16$ parallel steps each, i.e., 16 tokens are decoded per rollout step for efficiency. For the detailed hyperparameters, see Appendix C.

#### 7.3.1. MAIN RESULTS

Table 1 reports results on two LLaDA models across six reasoning benchmarks under a unified decoding regime, where each step decodes one token. The dataset-specific setups are provided in Table 7. We categorize methods into two groups: (i) *single-particle baselines*, which operate under a standard budget with a mean runtime of approximately 11.5s per sample. To ensure a fair assessment of latency, we measure runtime on a single A800 GPU by fixing the

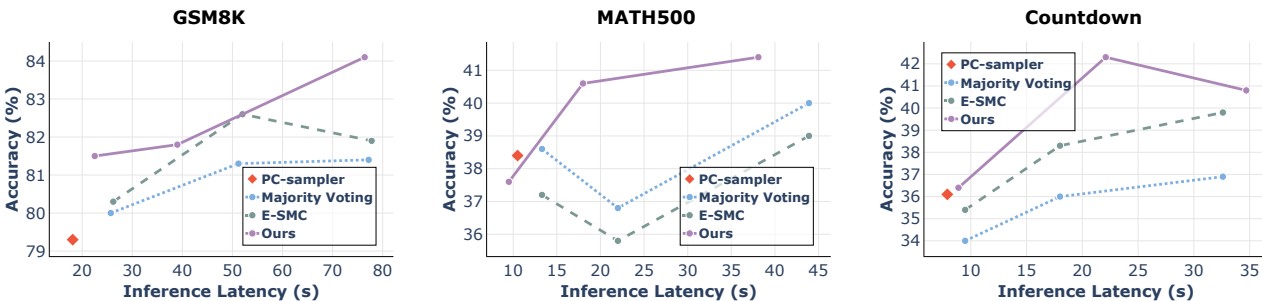

Figure 4. **Inference-time scaling efficiency on GSM8K, MATH500, and Countdown.** Accuracy is plotted against average inference latency (s) per sample. Our method (purple) consistently establishes a superior Pareto frontier across all tasks, scaling more effectively than Majority Voting (blue) and E-SMC (green).

prompt length to the median of each respective dataset and averaging results across all benchmarks. Among these, PC-Sampler consistently emerges as the strongest baseline. and (ii) *inference-scaling methods*, which explicitly trade extra computation for improved performance through multiple particles or iterative refinement. To maintain a rigorous budget-matched comparison, we report the best performance for each method under a fixed scaling budget. Specifically, POKE-SMC ($M = 4$) requires 42.2s per sample, representing a roughly $4\times$ overhead. We compare this against Majority Voting and E-SMC ($M = 5$), which require a nearly identical 42.5s.

**Scaling Efficiency Analysis.** By effectively utilizing the additional computation, POKE-SMC achieves average accuracy gains of 2.9% on LLaDA-8B-Instruct and 2.1% on LLaDA-1.5, establishing a new state-of-the-art for dLLM decoding. These improvements are most pronounced in reasoning-intensive tasks such as GSM8K and Countdown, where accuracy increases by 4.5%–5%. In contrast, alternative scaling strategies yield significantly lower returns. E-SMC provides only marginal average gains over the base policy (+0.8% on LLaDA and +0.3% on LLaDA-1.5). On benchmarks such as GSM8K and MATH500, E-SMC's performance is occasionally inferior to simple Majority Voting, whereas POKE-SMC maintains a substantial lead across all tasks. Furthermore, we compare against ReMDM, a self-contained framework (w/o PC-sampler) that utilizes iterative correction ($2\times$ overhead; see Appendix D). Despite its dedicated remasking and self-correction phases, ReMDM frequently fails to outperform even basic single-particle baselines. These results confirm that POKE-guided search is a far more effective mechanism for translating increased compute into reasoning capabilities than either independent sample aggregation or unguided iterative refinement.

### 7.3.2. PERFORMANCE-EFFICIENCY TRADE-OFF

To investigate how different methods scale with computational resources, we evaluate the performance-efficiency trade-off on GSM8K, MATH500, and Countdown. We con-

trol the runtime by adjusting the number of particles $M$ and the decoding steps $T$ to observe the evolution of accuracy (See Appendix E.3). As illustrated in Figure 4, POKE-SMC consistently establishes a superior performance–efficiency Pareto frontier across all benchmarks.

On GSM8K, POKE-SMC scales monotonically from 81.5% at 22.5s to 84.1% at 76.4s, while Majority Voting quickly saturates and E-SMC exhibits performance degradation at higher budgets. On MATH500, E-SMC and Majority Voting show clear non-monotonicity (e.g., Majority Voting 38.6% $\rightarrow$ 36.8% $\rightarrow$ 40.0%), whereas POKE-SMC improves monotonically from 37.6% to 41.4% as computation increases. A similar trend is observed on Countdown. While Majority Voting saturates early (34.0% $\rightarrow$ 36.9%) and E-SMC increases more gradually (35.4% $\rightarrow$ 39.8%), POKE-SMC achieves a substantially higher peak accuracy of 42.3% at 22.1s and remains competitive at larger budgets (40.8% at 34.7s). Overall, these results demonstrate that POKE-SMC not only utilizes additional computation more effectively, but also provides a more robust and scalable inference-time optimization strategy for complex reasoning tasks.

### 7.3.3. ABLATION STUDIES

To further dissect the factors contributing to the efficacy of POKE-SMC, we analyze the necessity of intermediate guidance and the robustness of our estimator in accelerated decoding scenarios.

We first investigate whether the gains of POKE-SMC stem solely from the discriminative power of the Path LL or from the active resampling mechanism. Table 2 compares our method against a **Best-of-$N$** strategy, which independently generates $M$ paths and selects the best one using the final Path LL. While Best-of-$N$ improves upon the base sampler, confirming Path LL as a robust proxy for sequence quality, it lags behind POKE-SMC, which achieves an additional 1.2% average gain. These results confirm that POKE is most effective as a dynamic guider, and its ability to re-allocate search budget toward promising trajectories mid-generation

*Table 2.* **Dynamic Guidance vs. Post-hoc Selection (LLaDA-8B, $M = 4$).** While Best-of-$N$ (reranking with Path LL) improves over the base sampler, our SMC-based POKE-SMC yields superior results by actively steering trajectories toward high-value regions mid-generation.

| Methods | HEval | MBPP | GSM8K | MATH | Countdown | Sudoku | Avg. |
|---|---|---|---|---|---|---|---|
| PC-sampler | 42.7 | 46.6 | 79.2 | 38.4 | 36.3 | 25.8 | 44.8 |
| +Best-of-N | 41.5 | **48.7** | 83.4 | 40.4 | 38.8 | 26.2 | 46.5 |
| +POKE (Ours) | **45.1** | 47.1 | **84.1** | **41.0** | **40.8** | **27.8** | **47.7** |

is superior to simply picking the best finished sequence.

Moreover, we also demonstrates superior performance in *high-throughput decoding scenarios*. As shown in Table 3, when unmasking two tokens per step, POKE-SMC maintains a substantial lead, achieving average improvements of **5.4%** and **4.9%** on the two LLaDA models, respectively, highlighting the robustness of POKE (See Appendix B.1). Additional ablations, including further comparisons with LookUM (Lee et al., 2025) and hyperparameter sensitivity analyses, are provided in Appendix B.2 and C, respectively.

## 8. Related Work

**Diffusion Language Models.** Diffusion models (Sohl-Dickstein et al., 2015; Ho et al., 2020; Song et al., 2020), successful in continuous image modeling (Dhariwal & Nichol, 2021), have been recently extended to discrete text generation (Hoogeboom et al., 2021; Campbell et al., 2022; Lou et al., 2023; Liu et al., 2025b). As a general foundational framework, D3PM (Austin et al., 2021a) formulates diffusion over discrete spaces with a fixed forward Markov process and a learnable reverse transition. Among its variants, Masked Diffusion Models (MDMs), also known as absorbing-state diffusion, have become the predominant approach for their simplicity and strong empirical performance. Recent theoretical refinements in training objectives (Sahoo et al., 2024; Ou et al., 2024; Shi et al., 2024) have facilitated the scaling of diffusion language models, exemplified by LLaDA (Nie et al., 2025) and Dream (Ye et al., 2025), positioning them as viable alternatives to autoregressive models. Furthermore, Reinforcement Learning (RL) techniques are now being employed to enhance dLLMs, evolving from early token-level adaptations (Zhao et al., 2025; Yang et al., 2025) to sophisticated sequence-level optimization strategies (Zhu et al., 2025; Wang et al., 2025a).

**Inference Strategies for dLLMs.** Beyond architectural scaling, the efficacy of dLLMs relies heavily on inference strategies that balance generation quality, speed, and flexibility. Central to these developments—and most relevant to our work—is the design of the token unmasking strategies. Early heuristic approaches rely on local uncertainty metrics to determine unmasking order, such as highest maximum probability (Chang et al., 2022), lowest entropy (Ye et al.,

2025; Ben-Hamu et al., 2025), or largest top-k confidence gap (Kim et al., 2025). Other approaches introduce structural priors via blockwise semi-autoregressive mechanisms (Arriola et al., 2025; Nie et al., 2025) or by integrating positional biases with content-aware confidence scores (Huang et al., 2025a). More recently, learning-based methods have emerged to explicitly optimize decoding paths by training planners or position selection policies (Huang et al., 2025b; Peng et al., 2025). To further enhance flexibility, some frameworks allow for the correction of earlier decisions via remasking mechanisms (Peng et al., 2025; Wang et al., 2025b). While these works focus on quality and flexibility, a parallel line of research emphasizes inference throughput via advanced parallel decoding strategies (Park et al., 2024; Israel et al., 2025; Wu et al., 2025b; Liu et al., 2025c). Concurrently, Chen et al. (2025) and Lee et al. (2025) also utilize Sequential Monte Carlo (SMC) to optimize decoding trajectories, yet differ from our approach by relying on one-step entropy-based value estimators (see Appendix D).

## 9. Limitations and Future Work

We presented a path-aware inference-time scaling framework for dLLMs based on the Path LL objective and the POKE estimator, and demonstrated its effectiveness through the POKE-SMC algorithm. While our results show consistent accuracy gains with favorable accuracy–compute trade-offs, our empirical evaluation is currently limited to 8B-scale models, such as LLaDA and LLADA-1.5. An important direction for future work is to validate the proposed framework on larger and more capable dLLMs (e.g., LLaDA2.0 (Bie et al., 2025) ) to assess its scalability under stronger base models.

In addition, the resampling strategy in POKE-SMC is straightforward, relying on periodic resampling at fixed intervals. Our algorithm, however, defines Path LL over a global variable scope, enabling direct comparison of partial states at different decoding stages. This property suggests more flexible inference mechanisms, such as adaptive rollback and regeneration when the base policy drifts toward low-probability regions, potentially inducing new inference-time scaling regimes beyond monotonic forward decoding.

## Impact Statement

This paper presents work whose goal is to advance the field of Machine Learning. There are many potential societal consequences of our work, none which we feel must be specifically highlighted here.

## Acknowledgement

This work was supported in part by the National Science and Technology Major Project (2022ZD0114902). It was also funded in part by the National University of Singapore under its Start-up Grant (Award No: SUG-251RES250). This work was additionally supported by Ant Group.

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

# A. Details of POKE-SMC

POKE-SMC maintains $M$ candidate trajectories (particles) $\{\boldsymbol{x}_{\mathcal{O}_t}^{(m)}\}_{m=1}^M$ that evolve as $t$ decreases from $T$ to 1. By utilizing POKE as a principled guidance signal, the algorithm dynamically reallocates its computational budget toward high-potential paths while pruning those likely to result in incoherent sequences. The process is structured into three iterative stages:

**1. Path Proposal** At step $t$, for each particle $m$, a set of tokens $\boldsymbol{x}_{\mathcal{Q}_t}^{(m)}$ is sampled from the dLLM reverse kernel, $\boldsymbol{x}_{\mathcal{Q}_t}^{(m)} \sim p_{\mathrm{D}}(\cdot \mid \boldsymbol{x}_{\mathcal{O}_t}^{(m)}; \theta)$, and update $\boldsymbol{x}_{\mathcal{O}_{t-1}}^{(m)} = \boldsymbol{x}_{\mathcal{O}_t}^{(m)} \cup \boldsymbol{x}_{\mathcal{Q}_t}^{(m)}$. To maintain efficiency, the proposal step can be performed for multiple iterations before triggering the evaluation stage. We define a resampling interval $\Delta$, such that weighting and resampling only occur when $(T - t + 1) \pmod \Delta = 0$.

**2. Global-Guided Weighting** To accurately compare particles that may follow different unmasking orders, we define importance weights based on the estimated total path log-likelihood. This ensures that all particles are evaluated on a uniform global scale regardless of which specific tokens they have unmasked so far. For a particle $m$ at a resampling step $t$, we first compute its accumulated path LL:

$$L_t^{(m)} = \sum_{j=T}^{t} \log p_{\mathrm{D}}(\boldsymbol{x}_{\mathcal{Q}_j}^{(m)} \mid \boldsymbol{x}_{\mathcal{O}_j}^{(m)}; \theta). \tag{14}$$

The total importance weight $w_t^{(m)}$ is constructed by augmenting this partial reward with $\hat{V}_{\mathrm{POKE}}$ that accounts for the future path LL:

$$\log w_t^{(m)} = \left( L_t^{(m)} + \hat{V}_{\mathrm{POKE}}(\boldsymbol{x}_{\mathcal{O}_{t-1}}^{(m)}) \right). \tag{15}$$

Here, $L_t^{(m)} + \hat{V}_{\mathrm{POKE}}$ represents a high-fidelity estimate of the final log-likelihood of the complete sequence.

**3. Resampling Step** To focus computational resources on the most promising trajectories, we perform multinomial resampling, where we introduce a resampling temperature $\tau_r > 0$ to control the sharpness of the selection process. Particles are sampled with replacement according to:

$$\Pr(\text{select particle } m) = \frac{\exp(\log w_t^{(m)}/\tau_r)}{\sum_{j=1}^{M} \exp(\log w_t^{(j)}/\tau_r)}. \tag{16}$$

Following resampling, the $M$ selected particles continue to the next proposal step. This mechanism effectively steers the decoder away from myopic, locally certain paths that lack global coherence. After $T$ steps, the sequence $\boldsymbol{x}^{(m)}$ from the particle with the highest terminal path LL is returned. The full pipeline is detailed in Algorithm 2.

# B. Additional Experiments

## B.1. Enhanced Advantage in Parallel Decoding

Standard local decoding heuristics in dLLMs often suffer from severe performance degradation as the parallelization level increases, primarily due to the compounding effect of sub-optimal early commitments. To evaluate the robustness of POKE-SMC under accelerated inference schedules, we investigate its performance under two accelerated configurations: a moderate parallel setting (**2 tokens per step**) across all benchmarks, and an aggressive parallel setting (**4 tokens per step**) across four representative tasks.

### B.1.1. EVALUATION UNDER 2-TOKEN PARALLEL DECODING

We first evaluate POKE-SMC in the parallel decoding setting by unmasking 2 tokens per step across all six benchmarks. As summarized in Table 3, the performance gap between POKE-SMC and alternative scaling baselines expands compared to the standard 1-token sequential setting.

Our method maintains a substantial and consistent lead, achieving an average absolute improvement of **5.4%** on LLaDA-8B-Instruct and **4.9%** on LLaDA-1.5-8B over the standard PC-sampler baseline, while outperforming the search baseline

E-SMC despite using fewer particles ($M = 4$ vs. $M = 5$). This demonstrates that our POKE estimator remains highly reliable even when the model generates more tokens at each step, making it a robust choice for faster inference scenarios.

*Table 3.* **Performance under parallel decoding (2 tokens per step).** POKE-SMC significantly outperforms all baselines across both LLaDA and LLaDA-1.5 models, demonstrating high robustness to reduced decoding steps.

| Methods | HEval | MBPP | GSM8K | MATH | Count. | Sudo. | Avg. |
|---|---|---|---|---|---|---|---|
| **LLaDA-8B-Instruct** | | | | | | | |
| PC-sampler | 22.0 | 39.8 | 78.5 | 34.4 | 29.4 | 24.2 | 38.1 |
| +Majority Voting ($M = 5$) | - | - | 81.4 | 36.6 | 36.0 | - | - |
| +E-SMC ($M = 5$) | 25.6 | 39.1 | 81.2 | 35.8 | 38.3 | 24.0 | 40.7 2.6↑ |
| +POKE (Ours, $M = 4$) | **26.8** | **44.7** | **81.5** | **40.2** | **42.3** | **25.6** | **43.5 5.4↑** |
| **LLaDA-1.5-8B** | | | | | | | |
| PC-sampler | 23.8 | 42.9 | 78.1 | 37.2 | 28.2 | 25.6 | 39.3 |
| +Majority Voting ($M = 5$) | - | - | 82.4 | **41.2** | 29.3 | - | - |
| +E-SMC ($M = 5$) | 25.0 | 43.6 | 82.5 | 38.2 | 34.9 | 26.6 | 41.8 2.5↑ |
| +POKE (Ours, $M = 4$) | **31.1** | **46.4** | **84.0** | 39.4 | **36.6** | **27.8** | **44.2 4.9↑** |

### B.1.2. STRESS TESTING UNDER AGGRESSIVE 4-TOKEN PARALLEL DECODING

To further push the limits of parallel inference, we stress-test POKE-SMC in a more aggressive setting that unmasks **4 tokens per step**. This evaluation is performed across 4 representative reasoning tasks under 3 distinct base policies using the LLaDA-8B-Instruct model. Table 4 tracks the performance migration from the raw base policy sampler to its POKE-SMC-augmented counterpart.

*Table 4.* **Performance migration under aggressive parallel decoding (4 tokens per step).** The table shows the accuracy boost (Base Policy → **POKE-SMC**) achieved across various base samplers on LLaDA-8B-Instruct.

| Base Policy | Countdown | MBPP | GSM8K | MATH500 | Avg. Gain |
|---|---|---|---|---|---|
| SemiAR+Conf | 26.1 → **32.6** | 28.8 → **34.5** | 73.9 → **75.5** | 31.5 → **33.6** | **+4.0** |
| PC-Sampler | 26.9 → **36.4** | 27.4 → **36.5** | 74.4 → **82.3** | 24.6 → **33.8** | **+8.9** |
| SemiAR+Uniform | 8.4 → **19.8** | 22.2 → **30.7** | 54.1 → **63.4** | 19.8 → **21.8** | **+7.8** |

The empirical scaling patterns in Table 4 indicate that the absolute advantage of POKE-SMC becomes even **more pronounced** as the decoding process becomes highly parallel. Specifically, POKE-SMC yields an extraordinary **+8.9%** average accuracy surge when sitting on top of the PC-Sampler, and a massive **+7.8%** boost on SemiAR+Uniform.

### B.2. Comparison with LookUM

We further compare POKE-SMC with **LookUM** (Lee et al., 2025), another entropy-guided SMC decoding method. To faithfully reproduce the original LookUM setting, we evaluate on **LLaDA-8B-Instruct** and use the same **base policy** for both methods: **Semi-AR + Entropy**. Specifically, we partition the sequence into sequential blocks and select tokens within each block using entropy-based unmasking. Following common Semi-AR configurations (Nie et al., 2025), we set the block size to 32 for all benchmarks, except for Sudoku where we use a single block (i.e., 1-block Semi-AR) to match the structured nature of the output format. All dataset prompts and evaluation protocols follow Table 7.

We run both LookUM and POKE-SMC under the same decoding regime: **one token per step** and $M = 4$ **particles**. This ensures that the two methods differ primarily in the *guidance signal* used to rank and resample candidate paths:

*Table 5.* **Comparison with LookUM on LLaDA-8B-Instruct under the LookUM base policy.** Both methods use Semi-AR+Entropy as the base policy (block size 32, except Sudoku with 1 block), one token per step, and $M = 4$ particles. Dataset settings follow Table 7.

| Methods | HumanEval | MBPP | GSM8K | MATH500 | Countdown | Sudoku | Avg. |
|---|---|---|---|---|---|---|---|
| LookUM | 42.7 | 45.4 | 81.8 | 37.6 | 33.0 | 25.0 | 44.3 |
| POKE-SMC | 41.5 | 46.2 | 82.6 | 40.0 | 35.0 | 26.4 | 45.3 |

**Discussion.** Under the reproduced LookUM setting, POKE-SMC achieves a higher average score ($44.3 \rightarrow 45.3$). The gains are consistent on most benchmarks, suggesting that replacing entropy-based guidance with a path LL objective improves the effectiveness of SMC-based inference scaling even when the underlying unmasking policy is fixed to Semi-AR+Entropy.

### B.3. Generalization to Dream-Instruct Architecture

To verify that POKE-SMC is model-agnostic, we evaluate its transferability to **Dream-Instruct-7B** (Ye et al., 2025), which belongs to a different dLLM family with distinct pre-training dynamics from LLaDA. We apply POKE-SMC ($M = 4$) on top of Dream-Instruct's default confidence-based decoding base policy without any model fine-tuning.

As shown in Table 6, integrating POKE-SMC yields consistent improvements across all four reasoning and coding benchmarks, achieving a substantial average absolute gain of **+4.9%** (and up to **+10.0%** on MBPP). These results confirm that the benefits of our framework are not architecture-specific.

*Table 6.* **Performance scaling on Dream-Instruct-7B.** Integrating POKE-SMC ($M = 4$) onto the default confidence base policy yields consistent accuracy gains across all four benchmarks.

| Task | Dream-Instruct-7B | + POKE-SMC ($M = 4$) | Absolute Gain |
|---|---|---|---|
| GSM8K | 80.9 | 83.0 | +2.1 |
| MATH500 | 45.8 | 47.6 | +1.8 |
| MBPP | 43.6 | 53.6 | +10.0 |
| Countdown | 20.0 | 25.8 | +5.8 |
| **Avg.** | 47.6 | **52.5** | **+4.9** |

## C. Hyperparameters and Sensitivity

*Table 7.* **Dataset-specific hyperparameters for Table 1.** PC-Sampler scoring hyperparameters follow Huang et al. (2025a).

| Dataset | Few-shot | Gen. Len. $L$ | Steps $T$ | $\tau_p$ | top-$k$ | Interval $\Delta$ |
|---|---|---|---|---|---|---|
| GSM8K | 4 | 256 | 256 | 1.0 | 8 | 64 |
| MATH500 | 0 | 256 | 256 | 1.0 | 4 | 64 |
| HumanEval | 0 | 256 | 256 | 1.0 | 6 | 64 |
| MBPP | 0 | 128 | 128 | 1.0 | 4 | 32 |
| Countdown | 3 | 128 | 128 | 1.0 | 4 | 32 |
| Sudoku | 5 | 128 | 128 | 0.1 | 4 | 32 |

Unless otherwise specified, we set the token sampling temperature $= 0$. For the **position scoring** used in the path proposal phase, we follow PC-Sampler and keep all its hyperparameters identical to the original paper (Huang et al., 2025a).

For the path generator, we use a categorical distribution over the top-$k$ candidate positions: $P(i \mid \mathbf{x}_{\mathcal{O}}) \propto \exp(s_i/\tau_p)$. We set $\tau_p = 0.1$ for Sudoku and $\tau_p = 1.0$ for all other benchmarks. We found $\texttt{top-}k = 4$ works well across datasets; additionally, $\texttt{top-}k = 8$ for GSM8K and $\texttt{top-}k = 6$ for HumanEval yield the best results.

For the SMC framework, we resample at a fixed interval $\Delta = T/4$ (i.e., three resampling stages over $T$ steps) with resampling temperature $\tau_r = 0.1$.

**Sensitivity to $\tau_p$ and top-$k$** We report sensitivity of the path generator hyperparameters on representative benchmarks; similar trends are observed on other datasets. All results follow the same setting as the main experiments (Table 1).

### C.1. E-SMC Hyperparameters and Controlled Comparison

**E-SMC (Original vs. Our Path-Generator Hyperparameters).** In the main table, we report E-SMC results using the hyperparameters recommended in its original paper (Chen et al., 2025), which adopts a more conservative path-generation strategy (notably $\texttt{top-}k = 2$, $\tau_p = 0.1$). To isolate the effect of the guidance signal (entropy vs. path-LL), we additionally evaluate E-SMC under our path-generator hyperparameters on LLaDA-8B-Instruct.

*Table 8.* **Sensitivity to path proposal hyperparameters.** Moderate top-$k$ ($k = 4$) values are generally robust.

| Setting | MATH500 ($L$=256, $T$=256) | Countdown ($L$=128, $T$=128) | MBPP ($L$=128, $T$=128) |
|---|---|---|---|
| $k = 4, \tau_p = 1.0$ | 41.0 | 40.8 | 47.1 |
| $k = 6, \tau_p = 1.0$ | 41.2 | 40.9 | 46.5 |
| $k = 8, \tau_p = 1.0$ | 39.8 | 40.5 | – |
| $k = 6, \tau_p = 0.5$ | – | – | 46.7 |

*Table 9.* **Controlled comparison on LLaDA-8B-Instruct.** E-SMC benefits from our less conservative path generator, but remains consistently below POKE-SMC.

| Methods | HumanEval | MBPP | GSM8K | MATH500 | Countdown | Sudoku | Avg. |
|---|---|---|---|---|---|---|---|
| E-SMC (original) | 41.5 | 45.0 | 81.9 | 39.0 | 40.4 | 25.6 | 45.6 |
| E-SMC (ours hparams) | 43.2 | 45.9 | 82.2 | 37.6 | 41.8 | 26.0 | 46.1 |
| POKE-SMC (POKE) | 45.1 | 47.1 | 84.1 | 41.0 | 40.8 | 27.8 | 47.7 |

**Discussion.** Even with comparable exploration and identical SMC resampling mechanics, our method still consistently outperforms E-SMC. This aligns with our correlation analysis (Figure 1), where path LL is substantially better correlated with final task accuracy than path entropy, thereby predicting the superior scaling behavior of POKE-SMC.

## D. Details of Baselines

This section provides implementation details and conceptual clarifications for the most relevant baselines considered in our experiments, with a particular focus on their guidance signals and inference-time scaling mechanisms.

**PC-Sampler (Base Policy)** We adopt **PC-Sampler** (Huang et al., 2025a) as the base unmasking policy for all multi-particle methods, including POKE-SMC, E-SMC, and Majority Voting. PC-Sampler, recently rebranded as *UnCode* in the original paper, selects the next unmasking position by combining token-level confidence with an explicit positional prior. For consistency with prior dLLM literature, we retain the name **PC-Sampler** throughout this work.

Formally, for each masked position $i \in \mathcal{M}$, PC-Sampler assigns a score

$$s_i = -H(p_\theta(\cdot \mid \boldsymbol{x}_\mathcal{O}, i)) \cdot \exp(-\lambda i), \qquad (17)$$

where $H(\cdot)$ denotes the cross-entropy between the model's predictive distribution at position $i$ and a background token distribution, serving as a confidence measure. The exponential decay term introduces a positional bias that prioritizes earlier positions in the sequence. All hyperparameters and implementation details strictly follow the original paper (Huang et al., 2025a).

**E-SMC and LookUM: Entropy-Guided SMC Search** E-SMC (Chen et al., 2025) and **LookUM** (Lee et al., 2025) are two closely related methods that apply Sequential Monte Carlo (SMC) search to dLLM decoding. Both methods aim to improve generation quality by maintaining multiple partial paths and periodically resampling them according to an entropy-based guidance signal. Given a partial observation $\boldsymbol{x}_\mathcal{O}$, both E-SMC and LookUM estimate the future uncertainty of the remaining masked tokens via the expected token entropy:

$$\mathcal{H}_{\text{future}}(\boldsymbol{x}_\mathcal{O}) = \mathbb{E}_{i \sim \mathcal{M}} \left[ \mathcal{H}(p_\theta(\cdot \mid \boldsymbol{x}_\mathcal{O}, i)) \right]. \qquad (18)$$

This quantity is then used as a surrogate value function, favoring partial paths with lower anticipated future entropy. While effective in certain reasoning tasks, it conditions solely on the current observed state $\boldsymbol{x}_\mathcal{O}$ and averages uncertainty independently across future positions, failing to account for long-range interactions.

**Comparison to Our Path LL Guidance.** In contrast, POKE-SMC employs a path-level resampling weight defined as the sum of the realized log-likelihood of observed tokens and an explicit estimate of the *future joint log-likelihood*:

$\log p_\theta(\boldsymbol{x}_\mathcal{O}) + \mathbb{E}[\log p_\theta(\boldsymbol{x}_\mathcal{M} \mid \boldsymbol{x}_\mathcal{O})]$. This formulation is principled in that it directly approximates the terminal path log-likelihood, which is a powerful optimization target of decoding. Moreover, entropy-based methods define their guidance over the *remaining masked set* $\mathcal{M}$. Since different candidate paths may follow different unmasking orders, they can have different remaining sets, making entropy scores computed on different variable scopes, which may not be directly comparable. In contrast, our guidance targets the same terminal objective and remains comparable across candidates (even in different decoding stages) regardless of which subset of tokens they already decode.

**ReMDM: Remasking-Based Self-Correction** ReMDM (Wang et al., 2025b) improves dLLM decoding by introducing iterative *remasking and self-correction*. Instead of searching over multiple paths in parallel, ReMDM repeatedly revisits previously generated tokens and selectively remasks low-confidence positions, allowing the model to revise earlier decisions.

While ReMDM scales computation by increasing the number of remasking iterations, its corrections remain *single-path* and are guided by local confidence heuristics rather than an explicit path-level objective. In our experiments, we set the number of remasking and correction steps equal to the generation length, corresponding to a $2\times$ computational overhead. Increasing the number of remasking rounds beyond this point does not yield further performance improvements.

## E. Experimental Details

### E.1. Details of Illustrative Experiments

**Settings for Figure 1 (Left)** The results are extracted from the same experimental runs as the LLADA-8B results in Table 1.

**Settings for Figure 1 (Right)** To investigate the relationship between sequence-level proxy scores and generation quality, we evaluate **LLADA** on the **GSM8K** dataset in a 0-shot setting. We set the generation length to 256 tokens, with a total of $T = 256$ steps and a block size of 32. For each problem, we generate samples with 3 ordering mechanisms:

- **Semi-AR + Random**: A semi-autoregressive order where tokens within each block of size 32 are generated in a random order.

- **AR**: A standard autoregressive order following a left-to-right generation.

- **Oracle-verified**: An order determined by a ground-truth (GT) verifier. Specifically, for each problem, we first sample 5 sequences with different orders; the verifier selects the first order that leads to a correct final answer. If none of the 5 samples are correct, we default to the first sampled order.

For each of these three orders, we sample 8 trajectories for each question in the dataset using a value temperature of 0.2 under a fixed order, resulting in 24 samples per question.

**Definitions of Baseline Proxy Scores** While we propose **Path LL** as our primary metric in the main text, we provide the formal definitions for the baseline proxies used for comparison here. The **ELBO** (Evidence Lower Bound) is calculated based on the training objective. For a given sequence $\boldsymbol{x}^0$, we estimate the ELBO using a Monte Carlo (MC) approach by averaging the loss over $K = 512$ random mask samples:

$$\text{ELBO}(\boldsymbol{x}^0) \approx \frac{1}{K} \sum_{k=1}^{K} \left[ \frac{1}{t_k} \log p_\text{D}(\boldsymbol{x}_{\mathcal{M}_{t_k}} \mid \boldsymbol{x}_{\mathcal{O}_{t_k}}; \theta) \right], \quad t_k \sim U(0, 1). \tag{19}$$

Additionally, we consider **Path Entropy** from Chen et al. (2025). First, the **State Entropy** ($h_{DE}$) is defined as the average Shannon entropy of the model's predictive distributions over the set of masked positions $\mathcal{M}_t$:

$$h_{DE}(t) = \frac{1}{|\mathcal{M}_t|} \sum_{i \in \mathcal{M}_t} \mathcal{H}[p_\theta(x_i \mid \boldsymbol{x}_{\mathcal{O}_t})]. \tag{20}$$

The **Path Entropy** is the integral of $h_{DE}$ over the generation time. In our discrete-time setting with $T$ steps, it is approximated by averaging the $h_{DE}$ across all steps:

$$\text{Path Entropy} = \frac{1}{T} \sum_{t=1}^{T} h_{DE}(t). \tag{21}$$

**Global Pooling Visualization**    In the main text, we focus on the per-question average accuracy curves. Here, we provide an additional version using **Global Pooling**. In this setting, rather than averaging curves across individual questions, we pool all samples from all questions together. We then rank this global pool of samples by their respective proxy scores and calculate the accuracy for each top-$k\%$ quantile. For both settings, we only consider questions whose accuracies fall between 10% and 90%, excluding trivial and unsolvable cases.

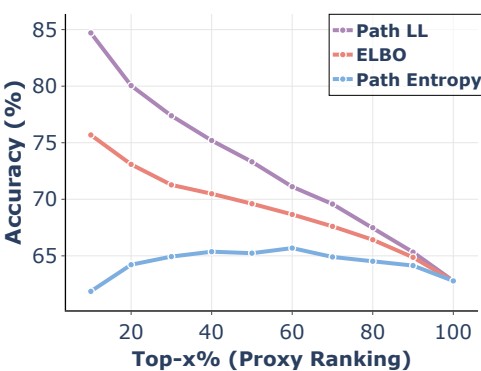

*Figure 5.* Comparison of Accuracy Curves (via Global Pooling) across Proxy Scores on GSM8K.

As illustrated in Figure 5, the trends observed in the global pooling curves are highly consistent with the per-question analysis. This indicates that the effectiveness of our proposed Path LL as a selection metric is robust.

### E.2. Details of Figure 2

To verify the effectiveness of future LL estimates with POKE, At **each** intermediate diffusion step, we assess the quality of the future LL estimates produced by **(i) POKE** and **(ii) product LL** $\sum q_\theta$. For a given step, we run lookahead with different values of $K$ (equivalently, different $|\mathcal{A}| = |\mathcal{M}|/K$) and obtain the future LL estimates. We then add this future LL estimates to the cumulative LL at that step to form an estimate of the final joint LL. These estimated joint LLs are finally compared against the ground-truth joint LL obtained after the full decoding process.

The experiment is conducted with LLaDA-8B-Instruct on a sample from the GSM8K dataset in a zero-shot setting. Generation is run for 256 tokens over $T = 256$ diffusion steps with a block size of 32. For each lookahead configuration, we draw 8 future trajectories in parallel and obtain the average product LL. We consider $K = 8$ and $K = 64$ as the maximum lookahead horizon. Throughout, we use entropy-based remasking with a remasking temperature of 1.0, and a value sampling temperature of T=0.1.

**Further Experiments**    To assess whether our findings generalize beyond a single dataset, we repeat the above experiment on GSM8K (using a different question), MATH500, and Countdown. We rerun the study with identical settings but finer-grained horizons $k \in \{4, 8, 16\}$, reporting results in Figures 6, 7 and 8. Across all datasets, POKE consistently yields estimates that are closer to the true final path LL than the product LL baseline, with the improvement most pronounced for small $K = 4$ and performs best at $K = 16$ (i.e., $|\mathcal{A}| = 16$. This observation further justifies our choice for the main experiments. Moreover, the true joint LL associated with correct answers is typically higher than that of incorrect answers, suggesting that joint LL is a meaningful signal for distinguishing solution quality.

### E.3. Details of Figure 4

This section provides the detailed experimental settings and numerical results underlying the performance–efficiency trade-off curves shown in Figure 4. We report accuracy versus per-sample latency on three representative benchmarks: GSM8K, MATH500, and Countdown. Latency is measured on a single NVIDIA A800 GPU, with prompt lengths fixed to the dataset median.

**Numerical Results.**    Tables 11, 12, and 10 list the exact accuracy–latency pairs used to generate the curves in Figure 4.

| Method | Latency (s) | Accuracy (%) |
|---|---|---|
| PC-Sampler | 18.1 | 79.3 |
| Majority Voting | 25.7 / 51.2 / 77.2 | 80.0 / 81.3 / 81.4 |
| E-SMC | 26.2 / 52.0 / 77.8 | 80.3 / 82.6 / 81.9 |
| POKE-SMC | 22.5 / 39.0 / 76.4 | 81.5 / 81.8 / 84.1 |

*Table 10.* Accuracy–latency pairs on GSM8K used in Figure 4.

| Method | Latency (s) | Accuracy (%) |
|---|---|---|
| PC-Sampler | 10.5 | 38.4 |
| Majority Voting | 13.3 / 22.0 / 43.9 | 38.6 / 36.8 / 40.0 |
| E-SMC | 13.3 / 22.0 / 43.9 | 37.2 / 35.8 / 39.0 |
| POKE-SMC | 9.5 / 18.0 / 38.1 | 37.6 / 40.6 / 41.4 |

*Table 11.* Accuracy–latency pairs on MATH500 used in Figure 4.

**Budget Control.** We control inference-time budgets by adjusting the number of particles $M$ and the number of tokens unmasked per step. For POKE-SMC, we use $M=4$ for 1token-per-step decoding, and $M=\{2,4\}$ for 2tokens-per-step decoding. For Majority Voting and E-SMC, we use $M=5$ for 1token-per-step decoding, and $M=\{3,5\}$ for 2tokens-per-step decoding. These configurations are chosen to yield comparable wall-clock latency across methods at each budget level.

## F. Theoretical Results

**Proof of Equation 7** Following Definition 5.1, we expand the KL divergence and let $q(\boldsymbol{x}_{\mathcal{M}} \mid \boldsymbol{x}_{\mathcal{O}}) := \prod_{i \in \mathcal{M}} p(x_i \mid \boldsymbol{x}_{\mathcal{O}})$, we obtain the key value decomposition :

$$
\begin{aligned}
\mathrm{TC}(\mathbf{X}_{\mathcal{M}} \mid \boldsymbol{x}_{\mathcal{O}}) &= D_{\mathrm{KL}}\Big(p(\boldsymbol{x}_{\mathcal{M}} \mid \boldsymbol{x}_{\mathcal{O}}) \,\Big\|\, q(\boldsymbol{x}_{\mathcal{M}} \mid \boldsymbol{x}_{\mathcal{O}})\Big) \\
&= \mathbb{E}_{p(\boldsymbol{x}_{\mathcal{M}}|\boldsymbol{x}_{\mathcal{O}})}\left[\log \frac{p(\boldsymbol{x}_{\mathcal{M}} \mid \boldsymbol{x}_{\mathcal{O}})}{q(\boldsymbol{x}_{\mathcal{M}} \mid \boldsymbol{x}_{\mathcal{O}})}\right] \\
&= \mathbb{E}_p[\log p(\boldsymbol{x}_{\mathcal{M}} \mid \boldsymbol{x}_{\mathcal{O}})] - \mathbb{E}_p[\log q(\boldsymbol{x}_{\mathcal{M}} \mid \boldsymbol{x}_{\mathcal{O}})]
\end{aligned}
$$

$$
\mathbb{E}_p[\log p(\boldsymbol{x}_{\mathcal{M}} \mid \boldsymbol{x}_{\mathcal{O}})] = \mathbb{E}_p[\log q(\boldsymbol{x}_{\mathcal{M}} \mid \boldsymbol{x}_{\mathcal{O}})] + \mathrm{TC}(\mathbf{X}_{\mathcal{M}} \mid \boldsymbol{x}_{\mathcal{O}})
$$

$$
\underbrace{\mathbb{E}_p[\log p(\boldsymbol{x}_{\mathcal{M}} \mid \boldsymbol{x}_{\mathcal{O}})]}_{V(\boldsymbol{x}_{\mathcal{O}})} = \mathbb{E}_p\left[\log \underbrace{\prod_{i \in \mathcal{M}} p(x_i \mid \boldsymbol{x}_{\mathcal{O}})}_{\text{Product LL}}\right] + \mathrm{TC}(\mathbf{X}_{\mathcal{M}} \mid \boldsymbol{x}_{\mathcal{O}}),
$$

and the last line is exactly Equation 7.

### F.1. Proof of Lemma 5.2

The proof of Lemma 5.2 follows from (Anari et al., 2025) and is included here for completeness.

We first introduce the definitiion of mutual information.

**Definition F.1** (Mutual information). Let $\mathbf{X}, \mathbf{Y}$ be random variables. The mutual information between is defined as

$$
\mathcal{I}(\mathbf{X}; \mathbf{Y}) = D_{KL}\Big(p(\mathbf{X}, \mathbf{Y}) \,\Big\|\, p(\mathbf{X}) \times p(\mathbf{Y})\Big)
$$

Using the above definition, we can introduce the "potential function" as follows:

| Method | Latency (s) | Accuracy (%) |
|---|---|---|
| PC-Sampler | 7.9 | 36.1 |
| Majority Voting | 9.5 / 18.0 / 32.6 | 34.0 / 36.0 / 36.9 |
| E-SMC | 9.5 / 18.0 / 32.6 | 35.4 / 38.3 / 39.8 |
| POKE-SMC | 8.9 / 22.1 / 34.7 | 36.4 / 42.3 / 40.8 |

*Table 12.* Accuracy–latency pairs on Countdown used in Figure 4.

**Definition F.2** (Potential Function). For masked variables $\mathbf{X}_{\mathcal{M}}$, the potential function is defined as:

$$\phi(\mathbf{X}_{\mathcal{M}} \mid \boldsymbol{x}_{\mathcal{O}}) = \sum_{i \in \mathcal{M}} I(\mathbf{X}_i; \mathbf{X}_{\mathcal{M} \setminus i} \mid \boldsymbol{x}_{\mathcal{O}}),$$

where $I(\cdot; \cdot \mid \boldsymbol{x}_{\mathcal{O}})$ denotes the conditional mutual information given the partial context.

**Proposition F.3.** *The potential function provides a principled upper bound for the total correlation, in particular*

$$\mathrm{TC}(\mathbf{X}_{\mathcal{M}} \mid \boldsymbol{x}_{\mathcal{O}}) \leq \phi(\mathbf{X}_{\mathcal{M}} \mid \boldsymbol{x}_{\mathcal{O}})$$

*Proof.* We first note that on finite spaces, both total correlation and mutual information admit simple entropy-based representations. In particular,

$$\mathrm{TC}(X_1, \ldots, X_n) = \sum_{i=1}^{n} \mathcal{H}(X_i) - \mathcal{H}(X_1, \ldots, X_n),$$

while for two random variables $X$ and $Y$, the mutual information can be written as

$$\mathcal{I}(X; Y) = \mathcal{H}(X) - \mathcal{H}(X \mid Y) = \mathcal{H}(X) + \mathcal{H}(Y) - \mathcal{H}(X, Y).$$

Consequently, mutual information is a special case of total correlation with two random variables, since

$$\mathrm{TC}(X, Y) = \mathcal{I}(X; Y).$$

To prove that $\mathrm{TC}(\mathbf{X}_{\mathcal{M}} \mid \boldsymbol{x}_{\mathcal{O}}) \leq \phi(\mathbf{X}_{\mathcal{M}} \mid \boldsymbol{x}_{\mathcal{O}})$, we expand the potential function using the entropy-based definition of conditional mutual information:

$$\phi(\mathbf{X}_{\mathcal{M}} \mid \boldsymbol{x}_{\mathcal{O}}) = \sum_{i \in \mathcal{M}} \mathcal{I}(X_i; \mathbf{X}_{\mathcal{M} \setminus \{i\}} \mid \boldsymbol{x}_{\mathcal{O}})$$

$$= \sum_{i \in \mathcal{M}} \left[ \mathcal{H}(X_i \mid \boldsymbol{x}_{\mathcal{O}}) - \mathcal{H}(X_i \mid \mathbf{X}_{\mathcal{M} \setminus \{i\}}, \boldsymbol{x}_{\mathcal{O}}) \right].$$

Using the chain rule of entropy identity $\mathcal{H}(X_i \mid \mathbf{X}_{\mathcal{M} \setminus \{i\}}, \boldsymbol{x}_{\mathcal{O}}) = \mathcal{H}(\mathbf{X}_{\mathcal{M}} \mid \boldsymbol{x}_{\mathcal{O}}) - \mathcal{H}(\mathbf{X}_{\mathcal{M} \setminus \{i\}} \mid \boldsymbol{x}_{\mathcal{O}})$, we rewrite the sum:

$$\phi(\mathbf{X}_{\mathcal{M}} \mid \boldsymbol{x}_{\mathcal{O}}) = \sum_{i \in \mathcal{M}} \mathcal{H}(X_i \mid \boldsymbol{x}_{\mathcal{O}}) - \sum_{i \in \mathcal{M}} \left[ \mathcal{H}(\mathbf{X}_{\mathcal{M}} \mid \boldsymbol{x}_{\mathcal{O}}) - \mathcal{H}(\mathbf{X}_{\mathcal{M} \setminus \{i\}} \mid \boldsymbol{x}_{\mathcal{O}}) \right]$$

$$= \sum_{i \in \mathcal{M}} \mathcal{H}(X_i \mid \boldsymbol{x}_{\mathcal{O}}) - |\mathcal{M}| \mathcal{H}(\mathbf{X}_{\mathcal{M}} \mid \boldsymbol{x}_{\mathcal{O}}) + \sum_{i \in \mathcal{M}} \mathcal{H}(\mathbf{X}_{\mathcal{M} \setminus \{i\}} \mid \boldsymbol{x}_{\mathcal{O}}). \tag{22}$$

Recall the definition of Total Correlation:

$$\mathrm{TC}(\mathbf{X}_{\mathcal{M}} \mid \boldsymbol{x}_{\mathcal{O}}) = \sum_{i \in \mathcal{M}} \mathcal{H}(X_i \mid \boldsymbol{x}_{\mathcal{O}}) - \mathcal{H}(\mathbf{X}_{\mathcal{M}} \mid \boldsymbol{x}_{\mathcal{O}}). \tag{23}$$

Subtracting equation 23 from equation 22, we see that $\phi(\mathbf{X}_{\mathcal{M}} \mid \boldsymbol{x}_{\mathcal{O}}) \geq \mathrm{TC}(\mathbf{X}_{\mathcal{M}} \mid \boldsymbol{x}_{\mathcal{O}})$ if and only if:

$$\sum_{i \in \mathcal{M}} \mathcal{H}(\mathbf{X}_{\mathcal{M} \setminus \{i\}} \mid \boldsymbol{x}_{\mathcal{O}}) - (|\mathcal{M}| - 1) \mathcal{H}(\mathbf{X}_{\mathcal{M}} \mid \boldsymbol{x}_{\mathcal{O}}) \geq 0.$$

Let $n = |\mathcal{M}|$. We must prove the inequality:

$$(n-1)\mathcal{H}(\mathbf{X}_{\mathcal{M}} \mid \boldsymbol{x}_{\mathcal{O}}) \le \sum_{i \in \mathcal{M}} \mathcal{H}(\mathbf{X}_{\mathcal{M} \backslash \{i\}} \mid \boldsymbol{x}_{\mathcal{O}}).$$

We prove this by induction on $n$ for $n \ge 2$.

**Base Case** ($n = 2$): For $\mathcal{M} = \{1, 2\}$, the inequality is $\mathcal{H}(X_1, X_2 \mid \boldsymbol{x}_{\mathcal{O}}) \le \mathcal{H}(X_2 \mid \boldsymbol{x}_{\mathcal{O}}) + \mathcal{H}(X_1 \mid \boldsymbol{x}_{\mathcal{O}})$. This is equivalent to $\mathcal{H}(X_1 \mid \boldsymbol{x}_{\mathcal{O}}) + \mathcal{H}(X_2 \mid \boldsymbol{x}_{\mathcal{O}}) - \mathcal{H}(X_1, X_2 \mid \boldsymbol{x}_{\mathcal{O}}) = \mathcal{I}(X_1; X_2 \mid \boldsymbol{x}_{\mathcal{O}}) \ge 0$, which is true by definition of mutual information.

**Inductive Step:** Suppose the hypothesis is true for $n - 1 \ge 2$. Fix an arbitrary $i \in \mathcal{M}$. For $j \ne i$, let $Y_j = (X_j \mid X_i, \boldsymbol{x}_{\mathcal{O}})$. Applying the induction hypothesis to $\{Y_j \mid j \ne i\}$ gives:

$$(n-2)\mathcal{H}(\mathbf{X}_{\mathcal{M} \backslash \{i\}} \mid X_i, \boldsymbol{x}_{\mathcal{O}}) \le \sum_{j \ne i} \mathcal{H}(\mathbf{X}_{\mathcal{M} \backslash \{i,j\}} \mid X_i, \boldsymbol{x}_{\mathcal{O}}).$$

Using the identity $\mathcal{H}(U \mid V) = \mathcal{H}(U, V) - \mathcal{H}(V)$, this implies:

$$(n-2)\left[\mathcal{H}(\mathbf{X}_{\mathcal{M}} \mid \boldsymbol{x}_{\mathcal{O}}) - \mathcal{H}(X_i \mid \boldsymbol{x}_{\mathcal{O}})\right] \le \sum_{j \ne i} \left[\mathcal{H}(\mathbf{X}_{\mathcal{M} \backslash \{j\}} \mid \boldsymbol{x}_{\mathcal{O}}) - \mathcal{H}(X_i \mid \boldsymbol{x}_{\mathcal{O}})\right].$$

Rearranging terms yields:

$$(n-2)\mathcal{H}(\mathbf{X}_{\mathcal{M}} \mid \boldsymbol{x}_{\mathcal{O}}) + \mathcal{H}(X_i \mid \boldsymbol{x}_{\mathcal{O}}) \le \sum_{j \ne i} \mathcal{H}(\mathbf{X}_{\mathcal{M} \backslash \{j\}} \mid \boldsymbol{x}_{\mathcal{O}}).$$

Summing these inequalities over all $i \in \mathcal{M}$:

$$n(n-2)\mathcal{H}(\mathbf{X}_{\mathcal{M}} \mid \boldsymbol{x}_{\mathcal{O}}) + \sum_{i \in \mathcal{M}} \mathcal{H}(X_i \mid \boldsymbol{x}_{\mathcal{O}}) \le (n-1) \sum_{j \in \mathcal{M}} \mathcal{H}(\mathbf{X}_{\mathcal{M} \backslash \{j\}} \mid \boldsymbol{x}_{\mathcal{O}}).$$

Using the fact that $\sum_{i=1}^{n} \mathcal{H}(X_i \mid \boldsymbol{x}_{\mathcal{O}}) \ge \mathcal{H}(\mathbf{X}_{\mathcal{M}} \mid \boldsymbol{x}_{\mathcal{O}})$ (subadditivity of entropy), we have:

$$(n-1) \sum_{j \in \mathcal{M}} \mathcal{H}(\mathbf{X}_{\mathcal{M} \backslash \{j\}} \mid \boldsymbol{x}_{\mathcal{O}}) \ge n(n-2)\mathcal{H}(\mathbf{X}_{\mathcal{M}} \mid \boldsymbol{x}_{\mathcal{O}}) + \mathcal{H}(\mathbf{X}_{\mathcal{M}} \mid \boldsymbol{x}_{\mathcal{O}})$$

$$= (n-1)^2 \mathcal{H}(\mathbf{X}_{\mathcal{M}} \mid \boldsymbol{x}_{\mathcal{O}}).$$

Dividing by $(n-1)$ confirms the inequality, thereby concluding the proof. $\square$

To further ensure computational efficiency without requiring additional model queries, we further simplify this into a bound based on individual token entropies:

**Proposition F.4.**

$$\mathrm{TC}(\mathbf{X}_{\mathcal{M}} \mid \boldsymbol{x}_{\mathcal{O}}) \le \phi(\mathbf{X}_{\mathcal{M}} \mid \boldsymbol{x}_{\mathcal{O}}) \le \sum_{i \in \mathcal{M}} \mathcal{H}(\mathbf{X}_i \mid \boldsymbol{x}_{\mathcal{O}}),$$

*Proof.* We aim to show that the potential function is bounded by the sum of marginal conditional entropies:

$$\phi(\mathbf{X}_{\mathcal{M}} \mid \boldsymbol{x}_{\mathcal{O}}) \le \sum_{i \in \mathcal{M}} \mathcal{H}(X_i \mid \boldsymbol{x}_{\mathcal{O}}).$$

Recall the definition of the potential function in terms of conditional mutual information:

$$\phi(\mathbf{X}_{\mathcal{M}} \mid \boldsymbol{x}_{\mathcal{O}}) = \sum_{i \in \mathcal{M}} \mathcal{I}(X_i; \mathbf{X}_{\mathcal{M} \backslash \{i\}} \mid \boldsymbol{x}_{\mathcal{O}}).$$

Using the entropy-based expansion of mutual information, $\mathcal{I}(A; B \mid C) = \mathcal{H}(A \mid C) - \mathcal{H}(A \mid B, C)$, we rewrite the potential as:

$$\phi(\mathbf{X}_{\mathcal{M}} \mid \boldsymbol{x}_{\mathcal{O}}) = \sum_{i \in \mathcal{M}} \left[ \mathcal{H}(X_i \mid \boldsymbol{x}_{\mathcal{O}}) - \mathcal{H}(X_i \mid \mathbf{X}_{\mathcal{M} \setminus \{i\}}, \boldsymbol{x}_{\mathcal{O}}) \right].$$

Separating the sum, we obtain:

$$\phi(\mathbf{X}_{\mathcal{M}} \mid \boldsymbol{x}_{\mathcal{O}}) = \sum_{i \in \mathcal{M}} \mathcal{H}(X_i \mid \boldsymbol{x}_{\mathcal{O}}) - \underbrace{\sum_{i \in \mathcal{M}} \mathcal{H}(X_i \mid \mathbf{X}_{\mathcal{M} \setminus \{i\}}, \boldsymbol{x}_{\mathcal{O}})}_{\geq 0}.$$

Since discrete conditional entropy is always non-negative ($\mathcal{H}(\cdot \mid \cdot) \geq 0$), removing the second term strictly increases (or maintains) the value. Therefore:

$$\phi(\mathbf{X}_{\mathcal{M}} \mid \boldsymbol{x}_{\mathcal{O}}) \leq \sum_{i \in \mathcal{M}} \mathcal{H}(X_i \mid \boldsymbol{x}_{\mathcal{O}}).$$

$\square$

**Proof of Lemma 5.2 (Cumulative Total Correlation Bound)** We aim to bound the expected cumulative total correlation:

$$\mathbb{E}_{\sigma} \left[ \sum_{k=1}^{K} \mathrm{TC}(\mathbf{X}_{A_k} \mid \mathbf{X}_{\mathcal{O}_k}) \right] \leq \frac{1}{K} \sum_{i \in \mathcal{M}} \mathcal{H}(X_i \mid \boldsymbol{x}_{\mathcal{O}}).$$

**Step 1: Relating Total Correlation to the Potential Function**
From Proposition F.3 , we established that for any set of variables and context, the total correlation is upper-bounded by the potential function $\phi$. Applying this to each block $A_k$ conditioned on its specific context $\mathbf{X}_{\mathcal{O}_k}$ (which includes the fixed context $\boldsymbol{x}_{\mathcal{O}}$ and previous blocks):

$$\mathrm{TC}(\mathbf{X}_{A_k} \mid \mathbf{X}_{\mathcal{O}_k}) \leq \phi(\mathbf{X}_{A_k} \mid \mathbf{X}_{\mathcal{O}_k}).$$

Summing over $k$ and taking the expectation over $\sigma$:

$$\mathbb{E}_{\sigma} \left[ \sum_{k=1}^{K} \mathrm{TC}(\mathbf{X}_{A_k} \mid \mathbf{X}_{\mathcal{O}_k}) \right] \leq \mathbb{E}_{\sigma} \left[ \sum_{k=1}^{K} \phi(\mathbf{X}_{A_k} \mid \mathbf{X}_{\mathcal{O}_k}) \right]. \tag{24}$$

**Step 2: Applying the New Pinning Lemma**
We invoke the New Pinning Lemma from (Anari et al., 2025) (Lemma 13 in the original text), adapted to our notation.

**Lemma F.5** (New Pinning Lemma (Anari et al., 2025))**.** *Let $\sigma$ be a random permutation of the masked indices $\mathcal{M}$ partitioned into $K$ blocks $A_1, \ldots, A_K$. Let $\mathbf{X}_{\mathcal{O}_k} = \mathbf{X}_{A_1 \cup \cdots \cup A_{k-1}} \cup \boldsymbol{x}_{\mathcal{O}}$. Then:*

$$\mathbb{E}_{\sigma} \left[ \sum_{k=1}^{K} \phi(\mathbf{X}_{A_k} \mid \mathbf{X}_{\mathcal{O}_k}) \right] \leq \frac{1}{K} \phi(\mathbf{X}_{\mathcal{M}} \mid \boldsymbol{x}_{\mathcal{O}}).$$

*Proof.* We omit the proof and refer the reader to (Anari et al., 2025).

Applying Lemma F.5 to the RHS of Inequality equation 24, we obtain:

$$\mathbb{E}_{\sigma} \left[ \sum_{k=1}^{K} \mathrm{TC}(\mathbf{X}_{A_k} \mid \mathbf{X}_{\mathcal{O}_k}) \right] \leq \frac{1}{K} \phi(\mathbf{X}_{\mathcal{M}} \mid \boldsymbol{x}_{\mathcal{O}}). \tag{25}$$

**Step 3: Bounding with Marginal Entropies**
From Proposition F.4, we established that the potential function of the full set is upper-bounded by the sum of marginal conditional entropies:

$$\phi(\mathbf{X}_{\mathcal{M}} \mid \boldsymbol{x}_{\mathcal{O}}) \leq \sum_{i \in \mathcal{M}} \mathcal{H}(X_i \mid \boldsymbol{x}_{\mathcal{O}}).$$

Substituting this into Inequality equation 25 yields the final result:

$$\mathbb{E}_\sigma \left[ \sum_{k=1}^{K} \mathrm{TC}(\mathbf{X}_{A_k} \mid \mathbf{X}_{\mathcal{O}_k}) \right] \leq \frac{1}{K} \sum_{i \in \mathcal{M}} \mathcal{H}(X_i \mid \boldsymbol{x}_{\mathcal{O}}). \tag{26}$$

$\square$

## G. Ranking Fidelity and Bound Tightness of POKE

In this appendix, we present a large-scale, systematic evaluation regarding the estimation accuracy and ranking preservation capability of the POKE estimator. We focus on two critical dimensions: (i) **Ranking Fidelity**, which directly tests whether POKE can identify high-potential partial decoding trajectories during the search process, and (ii) **Bound Tightness**, which quantifies the reduction in relative estimation error achieved by our entropy-based correction term compared to naive factorized baselines.

### G.1. Large-Scale Ranking Fidelity Analysis

We conduct an evaluation over 3 representative reasoning tasks (GSM8K, MATH500, and Countdown), randomly selecting 400 problems from each dataset and using LLaDA. For each problem, we sample 5 partial trajectories and halt the decoding process exactly at the intermediate stage (where 50% of the sequence tokens have been unmasked). Each partial state is scored using either our proposed POKE estimator or the standard Product-LL baseline (which lacks the total correlation correction term).

To construct an authoritative reference score for comparison, we establish a completion-based *Oracle proxy* by performing full future rollouts via the PC-sampler from each partial state, averaging the final Path LL over 8 independent runs. These extensive rollouts also determine the ultimate correctness of the completion trajectory. **Crucially, this oracle baseline is exceptionally heavy computationally: even a single high-fidelity rollout requires approximately $\sim 10\times$ more computational cost than a single evaluation pass of POKE.**

We rank the 5 alternative partial trajectories based on each scorer's value estimate and report the resulting top-$k$ completion correctness in Table 13. Here, $|A|$ denotes the parallelization budget, representing the number of tokens decoded per step during the POKE evaluation.

*Table 13.* Top-$k$ completion correctness (%) evaluating ranking fidelity at the 50% intermediate decoding stage across 3 tasks. $|A|$ specifies the number of tokens decoded per step. The completion oracle is obtained via computationally intensive full rollouts ($\sim 10\times$ slower than POKE).

| Task | Scorer | $|A|$ | Top-1 | Top-2 | Top-3 |
|---|---|---|---|---|---|
| **GSM8K** | Oracle | – | 79.86 | 79.76 | 78.57 |
| | POKE | 16 | 79.17 | 78.87 | 78.77 |
| | POKE | 8 | 79.76 | 78.42 | 78.77 |
| | Baseline | 8 | 78.87 | 79.12 | 78.97 |
| **MATH500** | Oracle | – | 32.52 | 32.32 | 32.18 |
| | POKE | 16 | 33.54 | 32.83 | 32.59 |
| | POKE | 8 | 32.32 | 32.11 | 31.64 |
| | Baseline | 8 | 31.30 | 32.62 | 31.44 |
| **Countdown** | Oracle | – | 52.41 | 47.89 | 43.37 |
| | POKE | 16 | 48.18 | 46.37 | 38.96 |
| | POKE | 8 | 48.59 | 43.37 | 39.96 |
| | Baseline | 8 | 48.80 | 44.88 | 40.76 |

The empirical findings in Table 13 substantiate that POKE consistently aligns more closely with the expensive rollout oracle compared to the Product-LL baseline. This superior ranking alignment holds true even under highly accelerated evaluation routines where a larger block is unmasked in parallel ($|A| = 16$). Despite operating at a fractional cost ($> 10\times$ cheaper), POKE nearly recovers the exact oracle top-1 accuracy on GSM8K (79.17% vs. 79.86%) and even matches or

slightly surpasses it on MATH500 (33.54% vs. 32.52%). This emphasizes that POKE successfully preserves the global sequence-level ranking oracle signal with remarkable fidelity while introducing minimal computational overhead.

## G.2. Bound Tightness and Estimation Error

To explicitly examine the theoretical tightness guaranteed by Lemma 5.2, we measure the relative estimation error of POKE against the Product-LL baseline (which factorizes tokens independently without the correction term). Evaluations are executed at the intermediate stage (50% tokens decoded). The ground-truth reference path log-likelihood is derived by averaging 1 token/step completion rollouts, which serves as an exact oracle proxy and is $\sim 10\times$ more expensive than a POKE forward pass. The quantitative results are outlined in Table 14.

*Table 14.* Relative estimation error (%) and error reduction percentage at the 50% intermediate decoding stage across datasets. $|A|$ denotes the number of tokens decoded per step.

| Dataset | $|A|$ | POKE Error (%) | Baseline Error (%) | Error Reduction (%) |
|---|---|---|---|---|
| **GSM8K** | 16 | 76.1 | 221.1 | 65.6 |
| | 8 | 34.6 | 120.8 | 71.4 |
| **MATH500** | 16 | 64.0 | 166.1 | 61.5 |
| | 8 | 26.2 | 97.8 | 73.2 |
| **Countdown** | 16 | 100.7 | 168.6 | 40.2 |
| | 8 | 48.6 | 91.2 | 46.7 |

The empirical evidence shows that POKE dramatically suppresses the relative estimation error compared to the baseline across all configurations, delivering substantial error reductions ranging from **40.2% to 73.2%**. This significant variance reduction confirms that incorporating the total correlation correction term drastically improves estimation accuracy. Together with the ranking validation in Table 13, these insights verify the assertion of Lemma 5.2 in terms of both structural estimation precision and sequence-level ranking preservation.

## G.3. Theoretical Convergence and Pinning Dynamics

This exceptional empirical tightness is strongly aligned with our theoretical expectations under Lemma 5.2. Crucially, Lemma 5.2 bounds the *expected cumulative dependence effect* integrated over the distribution of all possible future rollout orders, rather than attempting to bound the worst-case dependence under a single, static sequence permutation.

While a single deterministic unmasking sequence can be loose or arbitrarily highly correlated in the worst case, such persistent dependencies are highly unlikely to remain grouped as unresolved variables under uniformly random future rollouts. Guided by the foundational pinning analysis of Anari et al. (2025), once a subset of highly correlated variables is randomly selected and pinned (i.e., observed) at an early rollout stage, their long-range mutual dependencies are immediately conditioned away rather than being repeatedly accumulated into subsequent stages. This rapid shattering of token dependencies induces the $1/K$-type contraction rate that underpins Lemma 5.2. Consequently, our bound remains exceptionally tight in practice without necessitating any restrictive weak-dependence assumptions on the underlying language distribution.

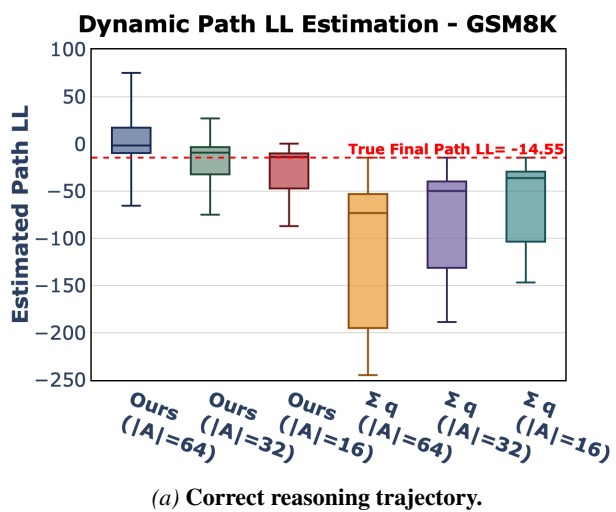
*(a)* **Correct reasoning trajectory.**

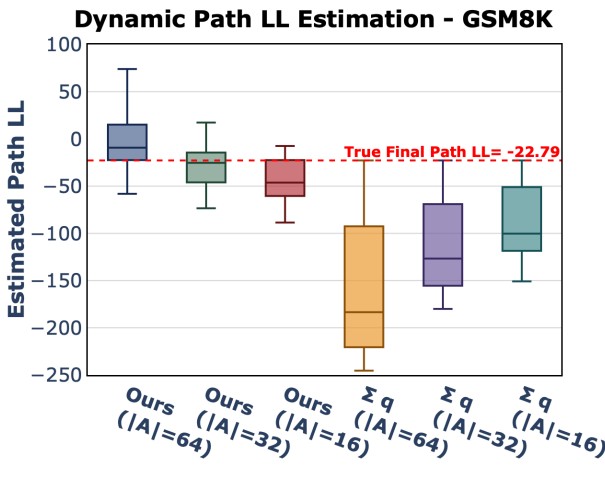
*(b)* **Incorrect reasoning trajectory.**

| Case: Mathematical Reasoning (GSM8K) | |
|---|---|
| **Prompt** | Please answer the question step by step and put the answer in \boxed{}. 
 Sunny is selling gingerbread and apple pie for a fundraiser. On Saturday, he sold 10 boxes of gingerbread and 4 fewer boxes of apple pie than on Sunday. On Sunday, he sold 5 more boxes of gingerbread than on Saturday and 15 boxes of apple pie. If the gingerbread cost \$6 and the apple pie cost \$15, how much did Sunny earn for two days? |
| **Generated (Correct)** | To determine how much Sunny earned for two days, we need to calculate the earnings for each day and then sum them up. 
 First, let's calculate the earnings for Saturday: - Sunny sold 10 boxes of gingerbread at 6 per box: $10 \times 6 = 60$ - He sold 11 boxes of apple pie at 15 per box: $11 \times 15 = 165$ - Total earnings for Saturday: $60 + 165 = 225$ 
 Next, let's calculate the earnings for Sunday: - Sunny sold 15 boxes of gingerbread at 6 per box: $15 \times 6 = 90$ - He sold 15 boxes of apple pie at 15 per box: $15 \times 15 = 225$ - Total earnings for Sunday: $90 + 225 = 315$ 
 Finally, we sum the earnings for both days: $225 + 315 = 540$ The answer is $\boxed{540}$ |
| **Generated (Incorrect)** | To determine how much Sunny earned for two days, we need to follow these steps: 
 1. Calculate the number of boxes of apple pie sold on Saturday. 2. Calculate the number of boxes of gingerbread sold on Sunday. 3. Calculate the earnings from gingerbread on both days. 4. Calculate the earnings from apple pie on both days. 5. Sum the earnings from both days. 
 1. On Saturday, Sunny sold 10 boxes of apple pie. 2. On Sunday, Sunny sold 15 boxes of apple pie. 3. On Saturday, Sunny sold 10 boxes of gingerbread. On Sunday, he sold $10 + 5 = 15$ boxes of gingerbread. 4. The earnings from gingerbread on both days is $10 \times 6 + 15 \times 6 = 60 + 90 = 150$. 5. The earnings from apple pie on both days is $15 \times 15 = 225$. 6. Therefore, the total earnings for the two days is $(150 + 225 = \boxed{375}$. |

*Figure 6.* Comparison of incorrect and correct reasoning trajectories on a GSM8K example. Despite identical prompts, premature commitment in the incorrect trajectory leads to an erroneous final answer, whereas coherent reasoning yields the correct solution.

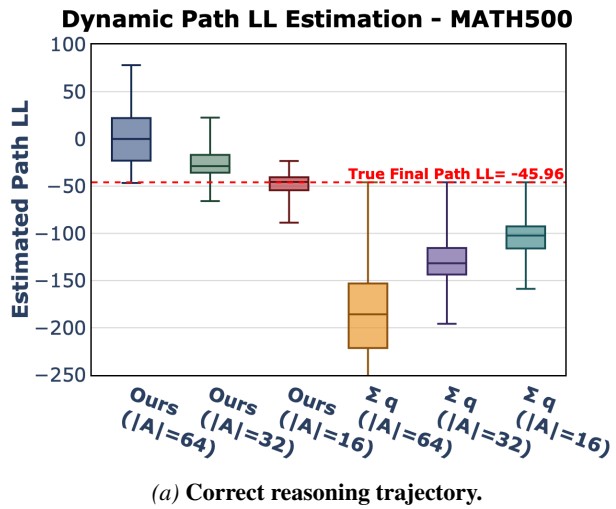

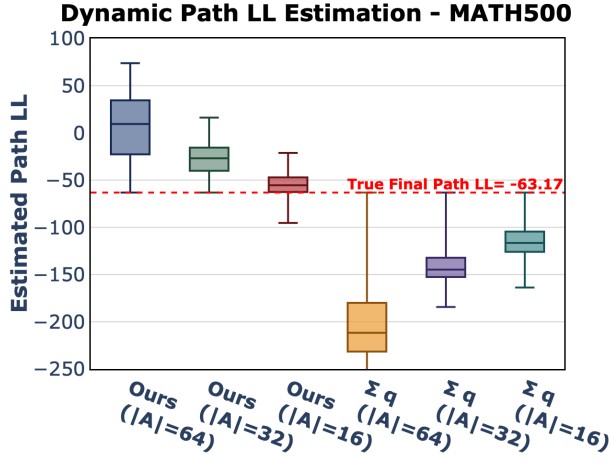

*(a)* **Correct reasoning trajectory.**          *(b)* **Incorrect reasoning trajectory.**

| **Case: Mathematical Reasoning (MATH500)** | |
|---|---|
| **Prompt** | Please answer the question step by step and put the answer in \boxed{}.
Consider the function

$$f(x) = \begin{cases} ax^2 & \text{if } x \geq a, \\ ax + 2a & \text{if } x < a \end{cases}$$

where $a$ is some number. What is the largest value of $a$ such that the graph of $y = f(x)$ intersects every horizontal line at least once? |
| **Generated (Correct)** | 1. To find the largest value of $a$ such that the graph of $y = f(x)$ intersects every horizontal line at least once, we need to analyze the behavior of the function $f(x)$.
2. For $x \geq a$, the function $f(x) = ax^2$. This is a parabola opening upwards, and it will intersect any horizontal line $y = k$ if $k \geq ax^2$.
3. For $x < a$, the function $f(x) = ax + 2a$. This is a linear function, and it will intersect any horizontal line $y = k$ if $k \leq ax + 2a$.
4. To ensure that every horizontal line intersects the graph at least once, $a$ must be such that $ax^2 \leq ax + 2a$.
5. Simplifying the inequality, we get $a \leq 2$.
Therefore, the largest value of $a$ such that the graph of $y = f(x)$ intersects every horizontal line at least once is $\boxed{2}$ |
| **Generated (Incorrect)** | 1. For $x \geq a$, the function $f(x) = ax^2$.
2. For $x < a$, the function $f(x) = ax + 2a$.
To find the largest value of $a$ such that the graph of $y = f(x)$ intersects every horizontal line at least once, we need to consider the behavior of the function.
- For $x \geq a$, $f(x) = ax^2$ is a parabola opening upwards. - For $x < a$, $f(x) = ax + 2a$ is a linear function.
To intersect every horizontal line at least once, the function must be continuous and have a sufficient range to cover all real numbers.
The function $f(x)$ is continuous for all $x$.
Thus, the largest value of $a$ that ensures the function intersects every horizontal line at least once is when $a$ is maximized.
Therefore, the largest value of $a$ such that the graph of $y = f(x)$ intersects every horizontal line at least once is $\boxed{1}$ |

*Figure 7.* Comparison of incorrect and correct reasoning trajectories on a MATH500 example. Despite identical prompts, premature commitment in the incorrect trajectory leads to an erroneous final answer, whereas coherent reasoning yields the correct solution.

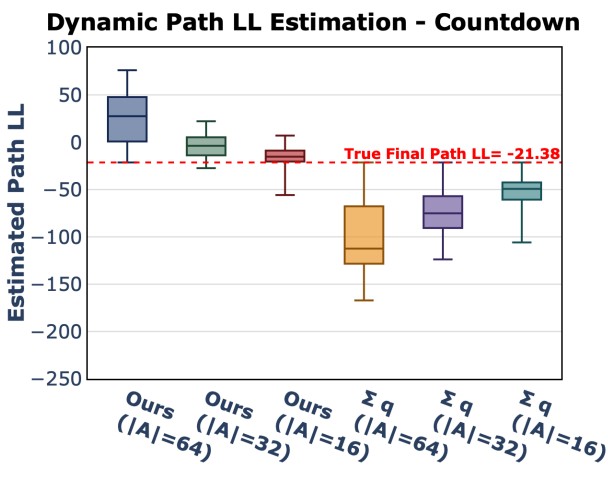

*(a)* **Correct reasoning trajectory.**

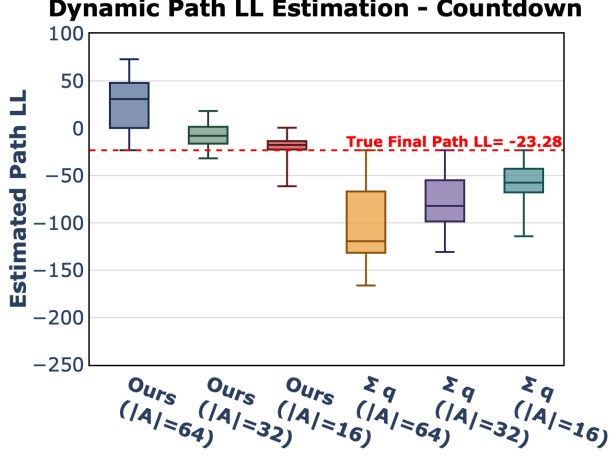

*(b)* **Incorrect reasoning trajectory.**

| Case: Arithmetic Reasoning (Countdown) | |
|---|---|
| **Prompt** | For the given numbers, find a sequence of arithmetic operations that results in the target number. Show your reasoning and conclude with "The answer is: [formula]". **Examples:** **Question:** 15,44,79,50 **Solution:** Let's try to combine 15 and 44. 44 - 15 = 29. Now we have 29 and the remaining number 79. We need to reach the target 50. Let's try 79 - 29 = 50. This works. The answer is: 44-15=29,79-29=50 **Question:** 1,2,12,25 **Solution:** We have 1, 2, 12 and the target is 25. Let's try multiplying 2 and 12. 2 * 12 = 24. Now we have 24 and the remaining number 1. We need to reach 25. 24 + 1 = 25. This is correct. The answer is: 2*12=24,1+24=25 **Question:** 3,85,5,30 **Solution:** The numbers are 3, 85, 5 and the target is 30.Let's try adding 85 and 5. 85 + 5 = 90. Now we have 90 and the remaining number 3. We need to reach 30. 90 / 3 = 30. That's the target. The answer is: 85+5=90,90/3=30 **New Question:** 86,28,13,31,96 |
| **Generated (Correct)** | The numbers are 86, 28, 13, 31, and the target is 96. Let's try adding 86 and 28. 86 + 28 = 114. Now we have 114 and the remaining numbers 13 and 31. We need to reach 96. Let's try adding 114 and 13. 114 + 13 = 127. Now we have 127 and the remaining number 31. We need to reach 96. Let's try adding 127 and 31. 127 + 31 = 158. Now we have 158 and the remaining number 96. We need to reach 96. Let's try subtracting 62 from 158. 158 - 62 = 96. This is correct. The answer is: 86+28=114,114+13=127,127+31-62=96 |
| **Generated (Incorrect)** | We have the numbers 86, 28, 13, 31, and the target is 96. Let's try adding 86 and 28. 86 + 28 = 114. Now we have 114 and the remaining numbers 13 and 31. We need to reach 96. Let's try adding 13 and 31. 13 + 31 = 44. Now we have 114 and the remaining number 44. We need to reach 96. Let's try subtracting 44 from 114. 114 - 44 = 70. Now we have 70 and the remaining number 26. We need to reach 96. Let's try adding 26 to 70. 70 + 26 = 96. This is correct. The answer is: 86+28=114,13+31=44,114-44=70,70+26 |

*Figure 8.* Comparison of incorrect and correct reasoning trajectories on a Countdown arithmetic reasoning example. Despite identical prompts, premature commitment and inconsistent bookkeeping in the incorrect trajectory lead to a spurious numerical prediction, whereas coherent multi-step reasoning yields the correct target value.

