# A. Details of POKE-SMC

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

9:     **end for**

10:    **// 2. Global-Guided Weighting & Resampling**

11:    **if** $(T - t + 1) \pmod \Delta = 0$ **or** $t = 1$ **then**

12:       **for each** particle $m$ **in parallel do**

13:          Compute $\hat{V}_{\mathrm{POKE}}(\mathbf{x}_{\mathcal{O}_{t-1}}^{(m)})$ via Algorithm 1

14:          $\log w_t^{(m)} \leftarrow L_{t-1}^{(m)} + \hat{V}_{\mathrm{POKE}}(\mathbf{x}_{\mathcal{O}_{t-1}}^{(m)})$

15:       **end for**

16:       Normalize: $\bar{w}_t^{(m)} \leftarrow \frac{\exp(\log w_t^{(m)}/\tau_r)}{\sum_{j=1}^M \exp(\log w_t^{(j)}/\tau_r)}$

17:       Resample $M$ particles from $\{\mathbf{x}_{\mathcal{O}_{t-1}}^{(m)}, L_{t-1}^{(m)}\}_{m=1}^M$ according to $\{\bar{w}_t^{(m)}\}_{m=1}^M$

18:    **end if**

19: **end for**

20: **Output**: Return $\boldsymbol{x}^{(m)}$ with the highest $L_0^{(m)}$.

---

block size to 32 for all benchmarks, except for Sudoku where we use a single block (i.e., 1-block Semi-AR) to match the structured nature of the output format. All dataset prompts and evaluation protocols follow Table 5.

We run both LookUM and POKE-SMC under the same decoding regime: **one token per step** and $M = 4$ **particles**. This ensures that the two methods differ primarily in the *guidance signal* used to rank and resample candidate paths:

**Discussion.** Under the reproduced LookUM setting, POKE-SMC achieves a higher average score ($44.3 \rightarrow 45.3$). The gains are consistent on most benchmarks, suggesting that replacing entropy-based guidance with a path LL objective improves the effectiveness of SMC-based inference scaling even when the underlying unmasking policy is fixed to Semi-AR+Entropy.

## C. Hyperparameters and Sensitivity

Unless otherwise specified, we set the token sampling temperature $= 0$. For the **position scoring** used in the path proposal phase, we follow PC-Sampler and keep all its hyperparameters identical to the original paper (Huang et al., 2025a).

For the path generator, we use a categorical distribution over the top-$k$ candidate positions: $P(i \mid \mathbf{x}_{\mathcal{O}}) \propto \exp(s_i/\tau_p)$. We set $\tau_p = 0.1$ for Sudoku and $\tau_p = 1.0$ for all other benchmarks. We found $\mathtt{top}\text{-}k = 4$ works well across datasets; additionally, $\mathtt{top}\text{-}k = 8$ for GSM8K and $\mathtt{top}\text{-}