# OpenReview forum: "Lookahead Path Likelihood Optimization for Diffusion LLMs"
_ICML.cc/2026/Conference — ICML 2026 regular_

### Official Review · Reviewer_z8qw · 2026-02-20

**Soundness:** 2
**Presentation:** 1
**Significance:** 2
**Originality:** 3
**Overall Recommendation:** 4
**Confidence:** 4

**Summary:**

This work proposes to estimate the future path Log-Likelihood to improve the decoding of dLLMs. It adopts the Monte Carlo-based search to implement the exploration and estimation. It looks effective based on the experimental results in this work.

**Compliance With Llm Reviewing Policy:**

Affirmed.

**Final Justification:**

The main concern has been addressed. I raise my score to 4 to appreciate the team's effort.

**Key Questions For Authors:**

Please refer to the weakness.

**Limitations:**

Yes.

**Strengths And Weaknesses:**

## Strength:

1. The insight that including the KL-divergence in path selection for dLLM is valuable.

2. This work explores optimization beyond AR models, which should be encouraged in the existing environment.

## Weakness:

1. The performance improvement over best-of-N is marginal (around 1%), and this trivial improvement comes at the cost of slower generation speed.

2. The evaluation is limited to the LLaDA architecture and should be extended to include Dream or more models.

3. The value function defined in Equation (11) is heuristic, and its validity has not been verified in the experiments.

4. The paper is hard to follow due to heavy notation. Lemma 5.2 has little contribution to the main purpose of this work.

---

> ### Author Rebuttal · Authors · 2026-03-31
>
> **Q1: The gain over Best-of-N is modest while introducing additional inference cost.**
>
> We thank the reviewer for this comment and would like to clarify the relationship between Best-of-N and our proposed POKE-SMC framework:
>
> 1. Best-of-N as a Justification for Path LL: In Table 2 of the paper, Best-of-N is implemented using our proposed Path LL as the reranking metric. The 1.7% improvement it achieves directly justifies the high fidelity of Path LL as a proxy for generation quality, which is one of our core contributions.
> 2. POKE-SMC as an Efficient Search Strategy with Better Trade-off: While Best-of-N uses Path LL only for post-hoc reranking, POKE-SMC leverages it dynamically during decoding to guide search, yielding a further ~1.2% gain over a strong Best-of-N baseline. Importantly, this improvement comes with only ~15% additional latency, resulting in a consistently better accuracy–latency trade-off (Fig. 3) compared to other inference-time scaling methods, indicating more effective use of extra compute.
>
> In conclusion, the ablation study shows that our contribution is two-fold: (1) a high-fidelity proxy (Path LL) and (2) an efficient search mechanism (POKE-SMC) that outperforms post-hoc selection at a marginal cost.
>
> **Q2: The evaluation is limited to LLaDA**
>
> Thank you for this suggestion. We additionally tested **Dream-Instruct-7B** on 4 representative tasks, using its default confidence-based decoding as the base policy and then applying POKE-SMC (M=4) on top. We observe consistent improvements across all four tasks, suggesting that the benefit of POKE-SMC is not specific to the LLaDA architecture, but transfers to another dLLM family as well:
>
> **T1: Dream Accuracy(%)**
> |Task|Dream|+POKE-SMC (M=4)|Gain|
> |-|-:|-:|-:|
> |GSM8K|80.9|83.0|+2.1|
> |MATH500|45.8|47.6|+1.8|
> |MBPP|43.6|53.6|+10.0|
> |Countdown|20.0|25.8|+5.8|
> |Avg.|47.6|52.5|+4.9|
>
> We will include these results in the revision to strengthen the claim that POKE-SMC is a model-agnostic inference-time scaling framework.
>
> **Q3: Whether Eq.(11) is a heuristic value function, and whether Lemma 5.2 meaningfully supports the main algorithm.**
>
> Thank you for this comment. We would like to clarify the intended logic of Section 5, and will make this derivation more explicit in the revision. As described around Eq. (9), we first decompose the true expected future Path LL into a tractable product-LL term plus a residual dependence term, namely the cumulative total correlation. Lemma 5.2 is then introduced precisely to control this residual under uniformly random future rollouts, by upper-bounding it with $\frac{1}{K}\sum_i H(X_i \mid x_O)$. Applying this bound to Eq. (9) directly yields Eq. (11), which is the POKE estimator used by the algorithm. In this sense, Eq. (11) is intended as a **theoretically motivated optimistic approximation** to the expected future Path LL.
>
> From this perspective, Lemma 5.2 is important to the main method: without it, the decomposition in Eq. (9) still contains an intractable dependence term, and we would not obtain the efficient estimator in Eq. (11). The key intuition, following the pinning analyses of Anari et al. [1,2], is that under random future revelation, long-range dependencies weaken in expectation, so a few-step parallel rollout can still provide a faithful estimate of future Path LL. We will revise Section 5 to make both this derivation and its intuition easier to follow.
>
> [1,2] Anari et al., Parallel Sampling via Counting (STOC 2024) and Parallel Sampling via Autospeculation (2025).
>
> **Q4: The validity of Eq.(11) has not been verified in the experiments.**
>
> We have provided qualitative evidence in Fig. 2 / Appx. G of the paper, and now extend this to a larger-scale evaluation on 3 tasks (400 problems each) using LLaDA. We measure the relative estimation error of POKE and the Product LL baseline (w/o the correction term) at an intermediate stage where 50% of the tokens have been decoded. The reference final LL is obtained by averaging 1-token-per-step completion rollouts, which we treat as **an oracle and is ~10× more expensive than POKE**. We report results for |A|=8,16, where (|A|) is the number of tokens decoded per step during POKE evaluation.
>
> **T2: Estimation Error(%)**
>
> |dataset|\|A\||POKE|Baseline|Error reduction|
> |-|-:|-:|-:|-:|
> |GSM8K|16|76.1|221.1|65.6|
> ||8|34.6|120.8|71.4|
> |MATH500|16|64.0|166.1|61.5|
> ||8|26.2|97.8|73.2|
> |Countdown|16|100.7|168.6|40.2|
> ||8|48.6|91.2|46.7|
>
> POKE consistently yields substantially smaller estimation error than Product LL, reducing error by 40%–73%, showing that the correction term systematically improves estimation fidelity.
>
> In addition, as reported in Table 4 of response to reviewer R7KR, we also perform a broad ranking analysis, where POKE’s top-k selection quality closely matches that of the same expensive oracle. Together, these results provide both quantitative estimation accuracy and ranking fidelity evidence supporting the practical validity of Eq. (11).

---

> > ### Author Rebuttal · Reviewer_z8qw · 2026-04-02
> >
> > The main concern has been resolved. Appreciate the team's effort.

---

### Official Review · Reviewer_R7KR · 2026-03-09

**Soundness:** 4
**Presentation:** 3
**Significance:** 3
**Originality:** 3
**Overall Recommendation:** 4
**Confidence:** 4

**Summary:**

A method about inference-time decoding for diffusion LLMs, where generation quality depends strongly on the unmasking order. The paper argues that commonly used heuristic orders are inconsistent across tasks, and proposes Path LL as a trajectory-conditioned objective that better correlates with downstream accuracy than ELBO- or entropy-based proxies. To optimize this objective at inference time, the authors introduce POKE, a lookahead value estimator for the expected future Path LL of a partial decoding state, decomposed into a product log-likelihood and a total-correlation correction term. They then integrate this estimator into a Sequential Monte Carlo decoding framework, resulting in POKE-SMC, which dynamically reallocates computation toward promising trajectories. Experiments on two LLaDA models across six reasoning-oriented benchmarks show consistent gains over heuristic baselines and entropy-guided inference-scaling baselines, with favorable accuracy-compute tradeoffs under matched budgets. :)

**Compliance With Llm Reviewing Policy:**

Affirmed.

**Final Justification:**

The main concern has been resolved. Good luck!

**Key Questions For Authors:**

1. POKE defines the state value under uniformly random future unmasking orders, while the actual decoding process uses a non-uniform proposal policy derived from PC-sampler scores and top-k sampling. Can the authors clarify why this mismatch is acceptable, and whether a rollout distribution closer to the actual proposal policy was considered? A stronger justification here would increase my confidence in the estimator design.
2. The current empirical evaluation is limited to two 8B-scale LLaDA-family models. Do the authors have any preliminary evidence, or at least a more concrete argument, that the same Path-LL-guided search framework should transfer to other dLLM families or stronger models? This would affect my view of the method’s broader significance.

**Limitations:**

The paper does acknowledge important technical limitations, including that the evaluation is currently restricted to 8B-scale LLaDA-family models and that the resampling strategy is relatively simple.

**Strengths And Weaknesses:**

## Strengths

- The introduction of Path LL is a meaningful conceptual contribution.
- The method is coherent and reasonably well motivated!
- Results are consistently positive.

## Weaknesses
- My main concern is about how fully the paper validates its central proxy. While the paper gives evidence that Path LL correlates better with accuracy than ELBO or path entropy, the most direct correlation analysis appears concentrated on GSM8K-style settings, with supporting evidence moved to the appendix. I would have liked a broader and more systematic validation that the ranking quality of Path LL generalizes across tasks and decoding regimes, especially since the paper’s main claim is fundamentally about Path LL being the “right” optimization target.
- A second concern is that POKE is presented as an optimistic estimator derived from an upper bound on cumulative total correlation. That makes intuitive sense for search, but the paper does not sufficiently quantify how loose this bound is in practice, nor how often the estimator misranks candidate partial trajectories relative to oracle final Path LL. The qualitative and small-scale fidelity plots are encouraging, but I would have preferred a more explicit large-scale ranking-fidelity analysis. Relatedly, the value function is defined under uniformly random future unmasking orders, whereas the actual search procedure uses a proposal policy based on PC-sampler scores plus top-k categorical sampling and periodic SMC resampling. The paper does not fully justify why value estimates under this rollout distribution should be especially well aligned with the actual search dynamics.

Overall, this is a good paper, and I am open to improving my score.

---

> ### Author Rebuttal · Authors · 2026-03-31
>
> **Q1: Does Path LL ranking generalize across tasks and decoding regimes?**
>
> We thank the reviewer for this question. We now extend the Path LL fidelity analysis to 4 tasks × 3 decoding regimes. Across settings, higher-Path-LL trajectories usually have higher accuracy, showing that the ranking quality of Path LL generalizes beyond GSM8K.
>
> For each task, we use 400 problems, sample 10 trajectories per problem, rank them by Path LL, and report the accuracy of the top p% trajectories.
>
> **T1: Semi-AR+conf (1 token/step)**
> |Task|p=10|p=30|p=50|p=70|p=90|p=100|
> |-|-:|-:|-:|-:|-:|-:|
> |GSM8K|80.0|78.3|77.4|77.0|76.5|76.1|
> |MATH500|38.9|38.1|37.9|38.0|38.3|38.1|
> |Countdown|39.9|38.2|36.1|33.3|30.4|29.2|
> |MBPP|46.9|46.4|46.0|45.7|45.0|44.7|
>
>
> **T2: Semi-AR+conf (2 tokens/step)**
> |Task|p=10|p=30|p=50|p=70|p=90|p=100|
> |-|-:|-:|-:|-:|-:|-:|
> |GSM8K|79.2|77.8|76.5|75.6|74.8|74.4|
> |MATH500|38.4|37.9|38.3|38.1|38.0|37.4|
> |Countdown|40.9|38.3|36.0|33.3|30.6|29.3|
> |MBPP|43.5|43.8|42.5|41.9|41.1|40.2|
>
> **T3: PC-sampler (1 token/step)**
> |Task|p=10|p=30|p=50|p=70|p=90|p=100|
> |-|-:|-:|-:|-:|-:|-:|
> |GSM8K|81.7|80.9|79.8|78.6|78.0|77.4|
> |MATH500|40.6|40.8|39.8|39.0|38.4|37.7|
> |Countdown|47.1|42.8|39.9|37.3|34.4|32.6|
> |MBPP|47.7|47.9|47.6|47.2|46.3|45.7|
>
> Overall, the trend is consistent across all 4 tasks and 3 regimes: higher Path LL generally implies higher accuracy. The pattern is strongest on GSM8K and Countdown, and remains visible on MBPP and MATH500 despite being noisier. This supports Path LL as a broadly useful ranking signal rather than a task- or regime-specific artifact.
>
> **Q2: I would have preferred a more explicit large-scale ranking-fidelity analysis.**
>
> We thank the reviewer for this important point. We now extend Fig. 2/Appx. G into a larger-scale ranking-fidelity analysis on 3 tasks (400 problems each) using LLaDA. For each problem, we sample 5 partial trajectories, stop at the halfway decoding stage (50% token are decoded), and score each partial state using either POKE or Product-LL (baseline w/o the correction term).
>
> To construct a strong completion-based reference, we perform full future rollouts using PC-sampler from each partial state and use the resulting final Path LL (averaged over 8 runs) as the completion score, with rollout completions also providing correctness. **We treat this as an oracle proxy, which is computationally expensive: even a single accurate rollout is ~10× more costly than POKE.**
>
> We rank the 5 partial trajectories using each scorer and report the resulting top-k completion correctness. This directly tests whether POKE preserves the oracle ranking signal. Here, |A| denotes the number of tokens decoded per step during POKE evaluation.
>
> **T4: Top-k completion correctness(%)**
> |Task|metric|\|A\||top-1|top-2|top-3|top-5|
> |---|---|---:|---:|---:|---:|---:|
> |GSM8K|Oracle||79.86|79.76|78.57|77.80|
> ||POKE|16|79.17|78.87|78.77|
> |||8|79.76|78.42|78.77|
> ||Baseline|8|78.87|79.12|78.97|
> |MATH500|Oracle||32.52|32.32|32.18|31.87|
> ||POKE|16|33.54|32.83|32.59|
> |||8|32.32|32.11|31.64|
> ||Baseline|8|31.30|32.62|31.44|
> |Countdown|Oracle||52.41|47.89|43.37|31.69|
> ||POKE|16|48.18|46.37|38.96|
> |||8|48.59|43.37|39.96|
> ||Baseline|8|48.80|44.88|40.76|
>
> POKE consistently aligns more closely with the oracle than Product-LL, even under faster rollouts (|A|=16). Despite being **>10× cheaper**, POKE closely matches the oracle ranking, for example, nearly recovering the oracle top-1 on GSM8K (79.17 vs. 79.86) and even surpassing it on MATH500 (33.5 vs. 32.5). This confirms that POKE preserves the oracle ranking signal with high fidelity while remaining highly efficient.
>
> For bound tightness, we also measure relative estimation error. At |A|=8, POKE reduces the error to 26.2%–48.6%, compared to 91.2%–120.8% for the baseline (see Table 3 of response to reviewer CB8o), confirming that POKE provides a practically tight approximation.
>
> **Q3: Why should value estimates under random rollouts align with the actual search dynamics?**
>
> The alignment is also supported by the above results. Importantly, our reference is not random completion, but a strong completion policy (1-token/step PC-sampler), i.e., the same proposal policy used in search. The fact that POKE, though defined via random future orders, remains close to this strong-policy reference suggests that it well captures the partial-state quality signal.
>
> Theoretically, random future orders are introduced for tractability and robustness (Sec. 4). Using the exact search rollout distribution would make value estimation much more expensive and intractable. Random rollouts instead provide a lightweight but informative estimate of future potential. This is also analogous to MCTS-style random simulation, where averaging over random paths can preserve relative ranking under mild conditions [1]. Our empirical results show that this ranking property transfers well to the actual search setting.
> [1] James et al. An Analysis of Monte Carlo Tree Search. AAAI, 2017.

---

> > ### Author Rebuttal · Reviewer_R7KR · 2026-04-01
> >
> > I thank the authors for the detailed rebuttal. The new cross-task Path LL fidelity analysis (Q1) and the large-scale ranking experiments (Q2) are convincing and have largely addressed my concerns. :)
> >
> > One point I would appreciate further clarification on: in the ranking-fidelity results for MATH500, POKE at |A|=16 achieves a top-1 accuracy of 33.54%, which actually surpasses the oracle reference (32.52%). Since the oracle is constructed from expensive full rollouts averaged over 8 runs, this inversion suggests the oracle proxy itself may carry non-trivial noise at this scale. Could the authors comment on the variance of the oracle estimates and what this implies for interpreting the ranking-fidelity results?
> >
> > This would help me better calibrate the strength of the evidence before finalizing my score.
> >
> > Thanks.

---

> > > ### Author Response · Authors · 2026-04-01
> > >
> > > We thank the reviewer for the positive feedback and for recognizing our additional analysis. Regarding the observed inversion on MATH500, we note that although we use a strong rollout policy and average over 8 rollouts to reduce variance, the oracle here does not represent the true optimal ordering. Therefore, it is not theoretically impossible to surpass it. While not noise-free, this procedure empirically yields a reasonably strong and stable oracle across datasets and top-k levels.
> > >
> > > Moreover, the inversion on MATH500 likely reflects dataset-specific noise. As shown in Table 3 (Q1), on MATH500, the top-30% accuracy (40.8) is slightly higher than the top-10% accuracy (40.6), suggesting a mild local inconsistency between Path LL and correctness on this dataset, which may also introduce small fluctuations in the results of Table 4. Overall, the oracle remains a reliable reference, and POKE maintains strong alignment with it.

---

### Official Review · Reviewer_FP6X · 2026-03-11

**Soundness:** 3
**Presentation:** 3
**Significance:** 3
**Originality:** 3
**Overall Recommendation:** 4
**Confidence:** 4

**Summary:**

This paper works on optimizing the decoding path of dLLMs by optimizing the path likelihood, an objective that involves a combination of product likelihood and a total correlation term. To efficiently estimate the objective at inference time, the author propose POKE which leverages dLLM's marginal entropy as an optimistic estimator to total correlation. Then, POKE is used to guide a Sequential Monte Carlo search algorithm for dLLM inference. Experiments on different dLLM inference tasks indicate the performance improvement of the proposed method.

**Compliance With Llm Reviewing Policy:**

Affirmed.

**Final Justification:**

The rebuttal addressed my main questions, and I will maintain the current score of weak accept.

**Key Questions For Authors:**

* In experiments, only 1-2 tokens are unmasked per step. I'm curious about applying your method to unmasking more tokens per step as fast sampling is one of the most promising advantage of dLLMs. Will its advantage over uniform/confidence become more significant?
* Your current motivation is based on the principle of admissibility in search, and optimism "prevents the search from prematurely pruning trajectories that, despite low local scores, retain high global potential". Can you provide more background on the connection between optimism and search algorithms? And would there be other theoretical motivation/justification for using optimism?
* I find it not easy to understand the SMC part in the paper. Although you cited the SMC paper but it would help if you provide an overview of SMC in the main text. What's the reason/motivation for using SMC? Are there other search algorithms than SMC which you can plug POKE into?

**Limitations:**

Yes

**Strengths And Weaknesses:**

**Strengths**:
* This paper works on improving the unmasking order of dLLMs and it is definitely a very important problem in this field.
* This paper provides sounded information-theoretic analysis on the optimistic estimator, providing deepened understanding of the path selection problem in dLLMs.
* The experimental evaluation of the proposed method is very comprehensive and the experiment results indicate improvement due to the proposed method.

**Weaknesses**:

See questions.

---

> ### Author Rebuttal · Authors · 2026-03-31
>
> **Q1: Will its advantage over uniform/confidence become more significant under more parallel decoding?**
>
> We additionally test a more aggressive parallel setting that unmasks **4 tokens per step** on 4 representative tasks under 3 base policies.
>
> | Base policy    |       Countdown |            MBPP |           GSM8K |         MATH500 | Avg. gain |
> | -------------- | --------------: | --------------: | --------------: | --------------: | --------: |
> | SemiAR+Conf    | 26.1 → **32.6** | 28.8 → **34.5** | 73.9 → **75.5** | 31.5 → **33.6** |  **+4.0** |
> | PC-Sampler     | 26.9 → **36.4** | 27.4 → **36.5** | 74.4 → **82.3** | 24.6 → **33.8** |  **+8.9** |
> | SemiAR+Uniform |  8.4 → **19.8** | 22.2 → **30.7** | 54.1 → **63.4** | 19.8 → **21.8** |  **+7.8** |
>
> Overall, the advantage of our method becomes **more pronounced** when more tokens are unmasked per step. As decoding becomes more parallel, local heuristics such as uniform/confidence make earlier and less reversible commitments, so trajectory-level lookahead becomes more valuable. This is also consistent with our current paper, where the gain already increases when moving from 1-token to 2-token decoding.
>
> Moreover, the gain varies across base policies: it is largest on the already strong PC-Sampler, followed by SemiAR+Uniform, and is smaller but still consistent on SemiAR+Conf.
>
> **Q2: Justification for Using Optimism**
>
> We thank the reviewer for this insightful comment. Our use of optimism is mainly connected to the classical notion of admissible heuristics in search, as exemplified by A* [1]. In A*, the heuristic estimates the remaining distance-to-go, and because the objective is to minimize cost, admissibility requires this estimate to be a lower bound on the true future cost. This lower-bounding property is what underlies A*’s optimality guarantee: the search does not discard a partial path that could still lead to the global optimum.
>
> Our setting is the maximization analogue of this idea. Since we aim to maximize future Path LL, optimism corresponds to using an upper bound on the achievable future value. While our full procedure does not enjoy the same exact optimality guarantee as classical A* due to additional approximations, the same intuition still applies: optimism reduces the risk of prematurely pruning a trajectory whose future global potential is not yet visible from its current local score.
>
> [1] Hart et al., 1968. "A Formal Basis for the Heuristic Determination of Minimum Cost Paths."
>
> **Q3: It would help if you provide an overview of SMC in the main text.**
>
> We thank the reviewer for this suggestion. The full procedural description is given in Appendix A and Algorithm 2. At a high level, for each prompt we maintain multiple partial decoding trajectories in parallel (e.g., 4 particles), each corresponding to a different partially generated sample under different unmasking orders. Starting from the fully masked sequence, we expand all particles for several decoding steps, then score each particle by its accumulated Path LL plus a POKE-based estimate of the remaining future Path LL (Eq. 11 and Algorithm 1). This score can be viewed as an estimate of the final trajectory quality if decoding continues from that state. We then resample, duplicating high-value particles and pruning low-value ones. By repeating this expand–evaluate–resample cycle, SMC progressively reallocates computation toward more promising decoding trajectories. We will incorporate this overview in the next version of the paper.
>
> **Q4: What's the motivation for using SMC? Are there other search algorithms than SMC which you can plug POKE into?**
>
> Our main motivation for using SMC is efficiency. In dLLMs, search must jointly reason over both token values and unmasking orders, which creates a much larger combinatorial search space than standard AR decoding. SMC is attractive in this setting because it keeps a fixed-size particle population and reallocates budget through resampling, avoiding the explicit tree expansion of methods such as MCTS while still supporting lookahead-based guidance. More broadly, POKE is not inherently tied to SMC: in principle, it could also be used as a value signal in other search procedures. We chose SMC because it provides a natural and tractable way to integrate trajectory-level lookahead into dLLM decoding.

---

> > ### Author Rebuttal · Reviewer_FP6X · 2026-04-02
> >
> > Thank you for your response. I don't have further questions.

---

### Official Review · Reviewer_CB8o · 2026-03-12

**Soundness:** 2
**Presentation:** 3
**Significance:** 2
**Originality:** 3
**Overall Recommendation:** 4
**Confidence:** 3

**Summary:**

The paper addresses the problem of optimizing the unmasking order during inference in dLLMs by proposing a path LL as a trajectory-aware objective that better correlates with downstream accuracy.

**Compliance With Llm Reviewing Policy:**

Affirmed.

**Final Justification:**

I am satisfied with the authors' final response and therefore have changed my score from weak reject to weak accept.

**Key Questions For Authors:**

1. My key concern is that a high path LL means the model is confident in this trajectory path. This includes two things: (a) the final sequence is genuinely good; (b) the model is overconfident about an **easy-but-wrong** completion. This paper does not seem to clarify this, which makes the good performance of POKE more like an engineering result.

2. Also, according to Lemma 5.2, the paper replaces the expected sum of total correlation with the sum of marginal entropy. This might be loose when tokens are highly correlated (high entropy), which makes POKE a poor discriminator between trajectories. The paper does not analyze the tightness of this bound empirically. Fig. 2 shows POKE tracks the oracle well, but only on a single GSM8K example; a more systematic analysis of the estimation gap across tasks would strengthen the paper.

3. The paper mentions MCTS but does not seriously compare against tree-search-based approaches. Given that MCTS has been successfully applied to LLM decoding, a discussion (or experiment) of why SMC is preferred over MCTS for dLLMs would be valuable. Are there fundamental structural reasons or primarily an engineering choice?

4. I’m not very familiar with the baselines in Fig. 3. Can the authors provide more explanations on each baseline and why POKE can achieve better latency over them?

**Limitations:**

See questions.

**Strengths And Weaknesses:**

I note two major strengths for this paper:
1. This motivation is clearly shown in Fig.1 that no single heuristic unmasking order dominates across tasks.
2. The paper is well-written, and the proposed method is evaluated on six diverse benchmarks.

---

> ### Author Rebuttal · Authors · 2026-03-31
>
> **Q1: Does Path LL reward wrong-but-confident paths?**
>
> While wrong-but-confident trajectories do exist, our results show that they are a minority, and Path LL remains a broadly faithful proxy for correctness across diverse tasks. Specifically, we sampled 10 trajectories per problem (400 problems each) across 4 representative tasks using LLaDA and evaluated:
> 1. Top-p% Accuracy: Accuracy within the top-ranked p% trajectories by Path LL.
> 2. Missed-Correct Rate@k (MCR@k): Probability that the top-k trajectories are all incorrect, given that at least one correct solution exists, which measures if wrong-but-confident paths overshadow correct ones.
>
> **T1: Top-p% Accuracy**
> |Task|p=10|p=30|p=50|p=70|p=90|p=100|
> |-|:-:|:-:|:-:|:-:|:-:|:-:|
> |GSM8K|80| 78.3 |77.4|77|76.5|76|
> |Countdown|40|38.2|36.1|33.3|30.4|29.2|
> |MBPP|46.9|46.4|46|45.7|45|44.7|
> |MATH500|0.39|38.4|37.9|38|38.3|38.1|
>
> **T2: MCR@k(%)**
> |Task|@1|@3|@5|@7|@9|
> |-|:-:|:-:|:-:|:-:|:-:|
> |GSM8K|10.1|6.3|3.8|1.9|1.1|
> |Countdown|6.9|5.2|3.9|3.1|0.9|
> |MBPP|7.3|3.9|1.9|0.9|0.5|
> |MATH500|24.1|19.4|14.2|8.4|2.9|
>
> The tables show a consistent pattern: Path LL is positively aligned with correctness, and overconfident misranking is limited. In Table 2, MCR@k decreases steadily as k increases, demonstrating that Path LL effectively ranks correct trajectories above incorrect ones. Therefore, Path LL provides a faithful enough signal to improve search in practice.
>
> **Q2: Tightness of Lemma 5.2**
>
> To assess the practical tightness of Lemma 5.2 more systematically, we extend the qualitative evidence in Fig. 2/Appx. G to a larger-scale analysis on 3 tasks (400 problems each) using LLaDA. We measure the relative estimation error of POKE and the Product LL baseline (w/o the correction term) at an intermediate stage where 50% of the tokens have been decoded. The reference final LL is obtained by averaging 1token/step completion rollouts, which we treat as **an oracle and is ~10× more expensive than POKE**. |A| denotes the #tokens decoded per step during POKE evaluation.
>
> **T3: Estimation Error(%)**
>
> |dataset|\|A\||POKE|Baseline|Error reduction|
> |-|-:|-:|-:|-:|
> |GSM8K|16|76.1|221.1|65.6|
> ||8|34.6|120.8|71.4|
> |MATH500|16|64.0|166.1|61.5|
> ||8|26.2|97.8|73.2|
> |Countdown|16|100.7|168.6|40.2|
> ||8|48.6|91.2|46.7|
>
> POKE significantly reduces estimation error over baseline (by 40%–73%). As further shown in Table 4 (response to reviewer R7KR), POKE’s ranking quality closely matches the much more expensive oracle. Together, these results validate Lemma 5.2 both in estimation accuracy and ranking fidelity.
>
> This is also theoretically expected. Lemma 5.2 controls the expected cumulative dependence effect over all future orders, rather than the dependence under any single rollout, which can be loose in the worst case. Under random rollouts, correlated variables are unlikely to remain grouped as unresolved variables across many future stages. Following the pinning analysis of Anari et al. [1,2], once some correlated variables are randomly pinned, the remaining dependence is conditioned away rather than repeatedly accumulated, yielding the 1/K-type contraction. Thus, the bound does not require weak dependence and instead leverages that random rollouts progressively break dependencies in expectation.
>
> [1,2] Anari et al., Parallel Sampling via Counting (2024) and Parallel Sampling via Autospeculation (2025).
>
> **Q3: Why SMC over MCTS in dLLMs**
>
> Compared to MCTS, SMC is better suited to dLLM decoding where the search space has a very large branching factor. Unlike AR LLMs where search only branches over next-token values, dLLMs must jointly reason over both token values and orders, making explicit tree expansion more intractable. In contrast, SMC (1) keeps a fixed number of particles rather than growing a tree, (2) reallocates computation by resampling toward more promising partial states, and (3) still supports trajectory-level lookahead evaluation.
>
> **Q4: Explanations of Fig.3 baselines and why POKE can achieve better latency**
>
> Fig. 3 includes a single-sample baseline, PC-Sampler, and three inference-scaling baselines built on top of it. PC-Sampler augments token confidence with an AR positional bias to improve performance (App. D). Majority Voting runs multiple independent rollouts and selects the final answer by self-consistency. E-SMC and POKE-SMC both use SMC: they maintain multiple partial decoding trajectories, periodically evaluate them, and resample to prune weaker paths to explore stronger ones (Alg. 2). The main difference is the guidance signal: E-SMC uses one-step entropy lookahead, while POKE estimates the expected future Path LL (App. D).
>
> POKE-SMC achieves a better accuracy–latency tradeoff for (1) it uses Path LL as a stronger optimization target, and (2) POKE enables efficient lookahead where future Path LL can be accurately estimated with fast rollouts. Therefore, it uses the same inference budget more effectively and converts extra compute into larger accuracy gains.

---

> > ### Author Rebuttal · Reviewer_CB8o · 2026-04-03
> >
> > Thank you for your detailed rebuttal and for taking the time to address my comments.
> >
> > The authors have responded to my main concerns and clarified their contributions.
> >
> > However, as the authors' table shows, the MCR@1 for MATH500 is **24.1%**,  meaning nearly a quarter of the time the top-ranked trajectory by Path LL is wrong despite correct solutions existing. That's a significant failure rate for a method whose entire search mechanism relies on Path LL as the ranking signal. Thus, I think without a principled mechanism to distinguish the easy-but-wrong cases, the method's reliability is task-dependent in ways the theory doesn't address.
> >
> > That said, I'll maintain my score.

---

> > > ### Author Response · Authors · 2026-04-03
> > >
> > > Thank you for the thoughtful follow-up and for clarifying your remaining concern. We would first like to clarify the interpretation of the MATH500 MCR@1 number. The reported 24.1% is computed on the subset of problems where at least one correct solution exists among the 10 sampled trajectories, i.e., the subset where pass@10 = 100%. Over the full evaluation set, the overall fraction of wrong-but-confident top-ranked cases is only **13.3%**, which is substantially smaller.
> > >
> > > To better contextualize the 24.1% figure, the meaningful practical reference is not pass@10, which assumes access to an oracle verifier, but the average per-sample accuracy of **66.8%**. In contrast, Path-LL-based top-1 selection achieves **75.9%** accuracy (1−24.1%), i.e., a **9.1-point improvement** using an achievable ranking signal, which is already a substantial gain.
> > >
> > > More fundamentally, we believe the key issue here is not whether Path LL is a perfect discriminator, but whether it is strong enough to guide search effectively without an oracle verifier. In this setting, no internal signal relying on model confidence can fully eliminate easy-but-wrong cases. However, **This limited failure rate does not undermine Path LL as a sufficiently faithful proxy for correctness to effectively improve decoding accuracy in practice. Table 1 directly supports this point**, showing that ranking by Path LL improves the top-10% accuracy over the average accuracy by **4.5 points** on average across the four tasks, with gains of up to **11 points**.
> > >
> > > Moreover, Path LL substantially outperforms prior internal-signal-based alternatives such as ELBO and path entropy in ranking quality, which further supports that it is not an arbitrary heuristic, but a more principled and practically effective target for guiding search.

---

### Decision · Program_Chairs · 2026-04-30

**Decision:**

Accept (regular)

**Comment:**

This paper proposes a principled framework for optimizing the unmasking order in Diffusion Large Language Models (dLLMs). It introduces Path LL (a trajectory aware objective that seems to correlate well with accuracy), POKE (an efficient value function estimator for future Path LL), and POKE-SMC (a Sequential Monte Carlo search approach). The reviewers agreed that optimizing generation order is an important challenge for dLLMs and found the conceptual introduction of Path LL to be well motivated. Initially, concerns were raised regarding whether Path LL might reward **wrong but confident** trajectories, the tightness of the POKE estimator's bounds, and the limited evaluation to only the LLaDA architecture. In the rebuttal, the authors provided a comprehensive rate analysis to show Path LL is a reasonable proxy, conducted large-scale experiments, and demonstrated generalization to another family of models: Dream-Instruct-7B. These additions satisfied the reviewers; Reviewer CB8o increased their score from a weak reject to a weak accept during the ac-reviewers discussion phase. Given the unanimous agreement among reviewers and the technical soundness of the results (including the newly added ones), the paper is recommended for acceptance.